

# Aerosol liquid water content in the moist southern West African monsoon layer and its radiative impact

Konrad Deetz[1], Heike Vogel[1], Sophie Haslett[2], Peter Knippertz[1], Hugh Coe[2], and Bernhard Vogel[1]

[1]Institute of Meteorology and Climate Research, Karlsruhe Institute of Technology (KIT), Karlsruhe, Germany
[2]National Centre for Atmospheric Science, and School of Earth and Environmental Sciences, University of Manchester, Manchester, United Kingdom

**Abstract.**

Water uptake can significantly increase the size and therefore alters the optical properties of aerosols. In this study, the regional-scale model framework COSMO-ART is applied to Southern West Africa (SWA) for a summer monsoon process study on 2–3 and 6–7 July 2016. The high moisture and aerosol burden in the monsoon layer makes SWA favorable to quantify properties that determine the aerosol liquid water content and its impact on radiative transfer. Given the marked diurnal cycle in SWA, the analysis is separated into three characteristic phases: (a) *Atlantic Inflow progression phase* (15-2 UTC), when winds from the Gulf of Guinea accelerate in the less turbulent evening and nighttime boundary layer, (b) *Moist morning phase* (3-8 UTC), when the passage of the Atlantic Inflow front leads to overall cool and moist conditions over land and (c) *Daytime drying phase* (9-15 UTC), in which the Atlantic Inflow front re-establishes with the inland heating initiated after sunrise. This diurnal cycle imprints, via the relative humidity, also the aerosol liquid water content. We analyzed the impact of relative humidity and clouds on the aerosol liquid water content. As shown by other studies, the accumulation mode particles are the dominant contributor of aerosol liquid water. We find aerosol growth factors of 2 (4) for submicron (coarse) mode particles, leading to a substantial increase of mean aerosol optical depth from 0.2 to 0.7. Considering the aerosol liquid water content leads to a decrease in shortwave radiation of about 20 W m$^{-2}$, while longwave effects appear to be insignificant, especially during nighttime. The estimated relationships between total column aerosol liquid water and radiation are -305±39 W g$^{-1}$ (shortwave in-cloud), -114±42 W g$^{-1}$ (shortwave off-cloud) and about -10 W g$^{-1}$ (longwave). The results highlight the need to consider the relative humidity dependency of aerosol optical depth in atmospheric models, particularly in moist tropical environments, where their effect on radiation can be very large.

## 1 Introduction

Water can significantly contribute to the total aerosol mass. While at low relative humidities (RH) the inorganic salts of an aerosol particle are solid, the particle spontaneously starts taking up water when exceeding a composition-specific RH, the *deliquescence relative humidity (DRE)* (Seinfeld and Pandis, 2006). The aerosol liquid water (ALWC) thereby affects the aerosol physicochemical and optical properties, which have the potential for significant impacts on the aerosol direct effect (ADE) (e.g. Jung et al., 2009).



The hygroscopic growth factor $GF$ is a frequently used measure to describe the aerosol hygroscopicity via the ratio between the wet aerosol diameter at a specific RH $d_{p,wet}(RH)$ and the dry aerosol diameter $d_{p,dry}$ (e.g. Chen et al., 2012). Furthermore, the relative ALWC is defined as the ratio between ALWC and the dry aerosol volume $V_{dry}$ to assess the mass of water that is taken up by a unit volume of dry aerosol (e.g. Bian et al., 2014).

In terms of ALWC, the understanding of aerosol effects on clouds and radiation is hindered by (a) the complexity in the reproduction of ALWC in observations and modeling under high RH (e.g. Bian et al., 2014) and (b) the covariance of cloud properties and aerosol water uptake with similar meteorological variables (e.g. RH; Andersen and Cermak, 2015). In atmospheric aerosol modeling the thermodynamic equilibrium model ISORROPIA II (Fountoukis and Nenes, 2007) is widely used (e.g. COSMO-ART, GEOS-Chem and LOTOS-EUROS), describing the chemical equilibrium between the gas, liquid and solid

phases for the aerosol system containing the inorganic ions potassium, magnesium, natrium, sulfate, nitrate, chloride and water. The ALWC is derived by using the Zdanovskii, Stokes and Robinson (ZSR) mixing rule (Stokes and Robinson, 1966). Hygroscopicity measurements can be obtained e.g. via the *High Humidity Tandem Differential Mobility Analyzer* (HH-TDMA, Hennig et al., 2005) for 90–98.5 % RH. Estimations of the aerosol hygroscopicity are also possible via the combination of observed aerosol number size distributions and aerosol compositions from aerosol mass spectrometers. Using the ZSR mixing

rule the $GF$ can be derived (e.g. Aklilu et al., 2006).

Several measurement campaigns and modeling efforts have addressed the ALWC and its impact on aerosol chemistry, visibility and the radiative transfer. The most comprehensive project in this regard is the *Haze in China (HaChi) campaign* in 2009. The scientific results are comprised in the ACP special issue "Haze in China (HaChi 2009–2010)". HaChi focused on the North China Plain between the megacities Beijing and Tianjin. The results indicate significant diurnal variations in the

aerosol physicochemical properties, including the aerosol scattering and absorption coefficient (high in the morning, low in the evening; Ma et al., 2011) and aerosol hygroscopicity (high during daytime, low during nighttime; Liu et al., 2011), both due to Planetary Boundary Layer (PBL) evolution and direct particle emissions. Chen et al. (2012) identify two haze regimes: below 90 % RH the haze is caused by high aerosol volume concentrations and above 90 % RH ALWC dominates the haze. Based on the HaChi observations, Kuang et al. (2015) conclude that the diurnal cycles of the optical properties single-scattering albedo

and asymmetry parameter differ when considering ambient or dry aerosol. For ambient aerosol, maximum (minimum) values are reached after sunrise (in late afternoon), correlated with the RH, whereas for dry aerosol, maximum values are detected at noon and minimum values in the morning and evening. Kuang et al. (2015) emphasize that the diurnal variations in the optical properties can significantly alter the ADE. Bian et al. (2014) estimate the maximum (average) value of ALWC in HaChi to be 971 (169) $\mu$g m$^{-3}$ related to a diurnal cycle with minimum values during day and maximum values during night. Due to

the high aerosol number and their hygroscopicity from aging and cloud processing, the ALWC contribution from the accumulation mode is dominating. For RH above 60 % the ALWC observations are in good agreement with the values derived from ISORROPIA II model. Liu et al. (2011) assess the hygroscopic properties of aerosol particles at high relative humidities and their diurnal variations in the North China Plain. They find average growth factors of 1.57-1.89 regarding dry diameters of 50-250 nm in a 95 % RH environment. For the highly hygroscopic particles a size increase by a factor of 2.1-2.8 (98.5 %

RH) compared to the dry diameter is reached. Liu et al. (2011) highlight that this behavior can significantly increase the light





scattering.

Aside from HaChi also the *Program of Regional Integrated Experiment of Air Quality over Pearl River Delta (PRIDE-PRD)* focused on the air pollution in China including aerosol hygroscopicity. The ALWC effect on the total light-extinction coefficient is estimated to be 34.2 %, with contributions from ammonium sulfate (25.8 %), ammonium nitrate (5.1 %) and sea salt (3.3 %) (Jung et al., 2009). Jung et al. (2009) highlight the sensitivity of the scattering and extinction coefficients as well as the mass-scattering efficiency and single-scattering albedo towards the ALWC. The modeling study of Cheng et al. (2008) for the same region reveals an aerosol-related cooling in the lower PBL, in which 40 % of the cooling effect is related to ALWC at 80 % RH.

The western Canadian aerosol study of Aklilu et al. (2006) reveals that the particle hygroscopicity is dominated by the availability of sulfate, since sulfate and $GF$ show significant correlations. Low $GF$ are detected for air masses affected by urban pollution. Aklilu et al. (2006) suggest that this is related to the primary organics that are less oxidized than secondary organics (e.g. Alfarra et al., 2004). Furthermore, Aklilu et al. (2006) underline the failing of the ZSR mixing rule for particulate nitrate that is subject to a considerably smaller water uptake than ammonium nitrate.

Andersen and Cermak (2015) analyze ten years of satellite-derived aerosol and cloud products and ERA-interim data over the Southeast Atlantic and find out that in very humid conditions (RH > 90 %) the RH increasingly affects the relationship between the amount of aerosol and the cloud droplet number concentration. However, Andersen and Cermak (2015) also stress that the biomass burning aerosol in this area is mostly situated above the cloud layer such that the boundary layer humidity might not be representative for the humidity in the aerosol layer.

Eastern China's tremendous air pollution makes it favorable for the study of ALWC, but a rapid growth of population and economy has also lead to a significant increase in atmospheric pollutants in Southern West Africa (SWA). Although SWA shows aerosol loadings similar to what is observed in East China (e.g. Hsu et al., 2017), the ALWC and its impacts on the visibility and radiative transfer have not been explored until now. Also SWA frequently shows a hazy milky sky even without the presence of clouds or the occurrence of a mineral dust event (personal observations by the authors), raising the question about the "Haze in SWA". First insights into West African ALWC characteristics are provided by observations obtained during the *African Monsoon multidisciplinary Analysis* (AMMA, Redelsperger et al., 2006). Matsuki et al. (2010) conclude that freshly emitted biomass burning aerosol is rather hydrophobic, whereas aging processes transform them into more hygroscopic particles. The aged biomass burning plumes, transported from Central to West Africa over the Atlantic Ocean (Mari et al., 2008), consist of highly hygroscopic particles with $GF > 1.2$ (Matsuki et al., 2010). Crumeyrolle et al. (2008) even see evidence for the coating of dust particles with soluble elements in Mesoscale Convective Systems enhancing their hygroscopicity and making them favorable as cloud condensation nuclei (CCN). However, the spatial focus of AMMA was on the Sahelian region and it is expected that the conditions farther south, over the coastal region of the Gulf of Guinea with its large urbanized areas and generally higher RH, differ substantially.

Furthermore, SWA is characterized by frequent nocturnal low-level stratus (NLLS) and stratocumulus (e.g. Schrage and Fink, 2012; Schuster et al., 2013; van der Linden et al., 2015; Adler et al., 2017) that have significant influence on the radiation budget (e.g. Hill et al., 2017). This study builds on the work of Deetz et al. (2018) that analyzes the impact of aerosol on the




properties of the Atlantic Inflow (AI) and Stratus-to-cumulus transition (SCT) by focusing on Ivory Coast. Deetz et al. (2018) highlight the dominance of the ADE and Twomey effect in the observed changes. The present study extends the aerosol impact analysis to the effects of ALWC on aerosol properties and the radiative transfer, because the model results analyzed in Deetz et al. (2018) reveal significantly enhanced Aerosol Optical Depth (AOD) in high RH regimes over SWA.

The goals of this study are: (1) to quantify the diurnal evolution of the ALWC-related properties and to assess whether diurnal recurring structures can be observed, which allow generalizing the results, (2) to evaluate the ALWC impact on radiative transfer, also in terms of their relevance to atmospheric modeling and (3) to derive robust relationships between ALWC and the change in radiative transfer. This also contributes to broaden the view of ALWC-radiation interactions to a highly polluted area other than Eastern China, an area that additionally is affected by the West African Monsoon (WAM) and its intense onshore

moisture transport.

This study is structured as follows: In Section 2 the model framework as well as the research area are introduced. The results comprise an analysis of atmospheric dynamics and thermodynamics affecting the ALWC (Sect. 3) and a detailed assessment of the radiative impact from ALWC (Sect. 4). The study concludes with a summary and evaluation of the findings (Sect. 5).

## 2   Model framework and setup

For this study, the regional-scale model framework COSMO-ART (Consortium for Small-scale Modeling - Aerosols and Reactive Trace gases, Vogel et al., 2009) is used. COSMO-ART is based on the operational weather forecast model COSMO (Baldauf et al., 2011) of the German Weather Service (DWD). The ART extensions allow for an online treatment of aerosol dynamics and atmospheric chemistry. This study accompanies the analysis of Deetz et al. (2018) using the same basic model setup, time period and spatial focus. The SWA model domain (2.5 km grid mesh size) comprises Ivory Coast, Ghana, Togo,

Benin and the Gulf of Guinea (red rectangles in Fig. 1). The SWA domain is nested into a coarse domain (blue rectangle in Fig. 1a) with a grid mesh size of 5 km to capture pollutants of mineral and biomass burning origin. The subsequent study will focus on the results of the red domain. The coarse domain is using ICON operational forecasts (approximately 13 km grid spacing) as meteorological boundary conditions. These cover the time period 25 June to 3 July to allow for an aerosol-chemistry spin up. The meteorological state is initialized every day at 0 UTC. COSMO-ART considers 12 lognormal aerosol

modes: (1) Aitken mode, (2) Aitken mode containing a soot core, (3) accumulation mode, (4) accumulation mode containing a soot core, (5) pure (fresh) soot, (6) coarse mode of anthropogenic origin, (7–10) coarse modes of marine origin (3 modes) and (11–13) coarse modes of mineral origin (3 modes). In the following, three aggregated modes are considered: AIT (Aitken mode, (1)+(2)), ACC (Accumulation mode, (3)+(4)) and COARSE (Coarse mode of marine origin, (7)+(8)+(9)). Pure soot as well as the coarse mode of anthropogenic and mineral origin are not considered, since their contribution to ALWC is either not

considered in COSMO-ART or the contribution is negligible. The model setup is summarized in Appendix A.

In COSMO/COSMO-ART the radiation scheme *General Radiative Algorithm Adapted to Linear-type Solutions radiation scheme* (GRAALS; Ritter and Geleyn, 1992) is used. Without the ART extensions, GRAALS considers aerosol climatologies (Tegen et al., 1997) instead of prognostic aerosol. In this case the aerosol is treated as dry and all effects emerging from the



ALWC are neglected (B. Ritter, personal communication, 2018). In contrast, COSMO-ART is able to derive the ALWC and its impact on the radiative transfer. With respect to anthropogenic aerosol the ALWC is calculated by ISORROPIA II (Fountoukis and Nenes, 2007). Fresh soot is treated separately but is also related to an uptake of water, namely via the condensation of sulfuric acid on the particle. Nevertheless, this contribution is negligibly small, since a soot particle with a considerable mass

fraction of sulfuric acid (more than 5 %) is shifted from the fresh soot mode to aged (internally mixed) aerosol treated by ISORROPIA II. Therefore we will not address to the ALWC from fresh soot in the subsequent analysis. In terms of sea salt, the ALWC is parameterized via Lundgren (2010). The coarse mode aerosols of anthropogenic and mineral origin are not related to ALWC in COSMO-ART and therefore also neglected in the following.

It has to be considered that activated aerosol particles are not removed from the aerosol distribution, which could lead to
potential double counts in the radiative transfer calculations. Prior approaches to remove the activated aerosol leads to a rapid and unrealistic cleaning of the atmosphere. With the model configuration denoted in Table A1, two realizations are performed: *Reference* considers ALWC in the calculation of the radiative transfer, whereas *No-ALWC* neglects this component. The difference between both realizations allows for a quantification of the changes in aerosol properties and their radiative effects in the monsoon layer.

The time period 2–3 July 2016 was selected due to the intense and persistent NLLS as observed at the Savè supersite (Kalthoff et al., 2018). Furthermore, 3 July is the center of the monsoon *Post-onset phase* (22 June – 20 July; Knippertz et al., 2017) and it is expected that the undistured monsoon condition favor and the process studies. Since the meteorological conditions show less variation from day to day, it is assumed that, even with a focus on a very short time period, insight can be achieved that can be generalized at least qualitatively to the length of the *Post-onset phase*. Nevertheless, we added results from the time period
6–7 July to assess the robustness of our findings.

## 3   Quantities affecting the ALWC

### 3.1   Impact of Atlantic Inflow (AI)

The studies of Adler et al. (2017) and Deetz et al. (2018) reveal a regular occurrence of the phenomenon *Atlantic Inflow (AI)* over SWA. AI involves a coastal front that develops during daytime and propagates inland in the evening. The AI front marks
the location of strongest horizontal gradients, with significantly higher wind speeds and lower potential temperatures post-frontally. The post-frontal area is affected by the nocturnal low-level jet (NLLJ) with a jet axis around 250 m Above Ground Layer (AGL). Furthermore, the post-frontal airmass is characterized by higher RH than the pre-frontal area. Therefore the AI frontal passage is characterized by a a decrease in temperature and an increase in RH and wind speed. By adapting the method proposed by Grams et al. (2010), Deetz et al. (2018) identify the front in the model output by the location at which a specific
isentrope of potential temperature $\theta_s$ (302 K) crosses a specific height $h_s$ (250 m AGL). Especially over the flat terrain in Ivory Coast a coherent coast-parallel frontal pattern can be observed. Therefore in the following the spatial focus is set to Ivory Coast.

AI is the underlying meteorological process affecting all measures relevant for ALWC, such as RH and the meridional transport



of specific aerosol types in the levels below 1 km AGL. It will be shown that the diurnal cycle of RH is not only thermodynamically (radiative cooling) but also dynamically driven (cold air advection via AI). In fact, it is a superposition of both effects, which are hard to disentangle. The AI impact will be included in the following analysis by taking into account the AI front locations. A detailed assessment of the spatiotemporal properties of AI and its aerosol sensitivity is presented in Deetz et al.

(2018), based on the same model setup and time period as used for this study. Subsequently, we focus on the time period 15 UTC to 15 UTC of the following day to capture a full diurnal cycle starting with the time of the inland propagation of the AI front. As described in Deetz et al. (2018), a decrease in surface heating leads to a deceleration of the inland propagation of the AI front. Although this mechanism is assessed in detail in Deetz et al. (2018) and will therefore not be in the focus of this study, we need to be aware of similar effects that could arise from changing the representation of ALWC in the radiative calculations.

An analysis of the frontal position reveals that the consideration of the ALWC leads to an AI front ahead of the *No-ALWC* front by about 6–7 km on 2–3 July between 15 UTC and 22 UTC (similar for 6–7 July). It is hypothesized that including the ALWC leads to a daytime inland cooling that reduces the turbulence over land and favors the earlier onshore propagation of the AI front. Possibly, also reduced nighttime longwave cooling due to ALWC can favor the persistence of the local heat low inland that accelerates the front. With respect to 2 July 22 UTC and by comparing with the findings of Deetz et al. (2018), the

displacement amplitude is 2.5 times smaller than the displacement from reducing the aerosol amount by a factor of 10. Since the frontal displacement is small and virtually constant in time, the following study will not further assess this aspect.

### 3.2 Impact of Relative Humidity (RH)

COSMO-ART shows reasonable results in numerous comparisons with other DACCIWA observations (e.g. Deetz, 2018; Deetz et al., 2018). Unfortunately, no aircraft observations are obtained on 3 and 4 July over Ivory Coast during the DACCIWA

campaign and the aircraft payload does not include devices to directly measure ALWC. For this study, radiosoundings of Lamto and Abidjan (Maranan and Fink, 2016) are used to compare observed RH vertical profiles with the COSMO-ART results. The intercomparison is presented in Appendix B. Generally, better agreement is achieved in the inland station Lamto (Fig. B1) than in the coastal station Abidjan (Fig. B2 and B3), likely due to the more complex boundary layer structure near the coast. The intercomparison show times with very good agreement (especialy during night on 3 July (Fig. B2g,h) and on

6 July at Lamto (Fig. B1a,b)). Interestingly, these times with good agreement simultaneously denote an agreement in the AI-related low-level moisture increase during night. Significant underestimations frequently occur approximately above 1000 m AGL (e.g. on 6 July 23 UTC at Abidjan (Fig. B3d)). In general, COSMO-ART is able to reasonably reproduce the RH vertical profile over Ivory Coast. The tendency of the model to underestimate RH implies that the model-derived ALWC is a lower limit that can actually be higher in the field.

Figure 2 presents first insight in the diurnal cycle of RH (Fig. 2a) and ALWC (Fig. 2b) via spatial mean vertical profiles in the monsoon layer over Ivory Coast. Generally moist conditions (RH > 70 %) can be observed over the area of interest (Fig. 2a). From 15 UTC to about 0 UTC the maximum RH is located at or above 1000 m AGL. For the layer below 750 m AGL a significant increase in RH is visible from 15 UTC (below 75 %) to 9 UTC on the next day (above 95%). In the following, it will be shown that this is related to the onshore advection of air with higher RH (colder air) within the AI. Highest RH on



the order of 95 % are reached between 3 UTC and 12 UTC in the lowest 1500 m AGL. This is reflected in the vertical profile of ALWC, showing highest values consistent with RH (Fig. 2b), with remarkable diurnal variations encompassing two orders of magnitude. Therefore it can be deduced that ALWC is most sensitive to the morning hours (sunrise in Abidjan is around 6 UTC (6 local time)). Based on Figure 2 we focus on the lowest 1500 m AGL in the following, capturing the monsoon layer.

The DACCIWA measurement campaign reveals that the monsoon layer over SWA shows typical heights of about 1900 m AGL (Kalthoff et al., 2018). As denoted in Section 2, the ALWC calculation for secondary inorganic particles in COSMO-ART is treated using ISORROPIA II. Bian et al. (2014) showed that robust results for the ALWC can be expected from ISORROPIA II for RH > 60 %. As presented in Figure 2, the average conditions over Ivory Coast reveal RH above 70 %. Therefore, principally we also can expect robust results for SWA. Figure 3 shows a Hovmöller diagram for the median RH (Fig. 3a) and total ALWC

(Fig. 3b) in the lowest 1500 m AGL as zonal means over Ivory Coast (7.5° W–3° W, 4–10° N) between 2 July 15 UTC and 3 July 15 UTC. The black bars denote the location of the 302 K isentrope at 250 m AGL that is used for the frontal detection between 15 UTC and 22 UTC (Deetz, 2018). In the first half of the presented time period a clear separation between the pre-frontal inland area (north of the black bars in Fig. 3a) with relatively low RH and the post-frontal area (south of the black bars in Fig. 3a) with relatively high RH can be observed. The inland propagation of the front after 2 July 15 UTC is related

to advection of cooler post-frontal air. In the following, this time period is denoted as Phase 1 (*AI progression phase*, 15–2 UTC). After the front has passed the area, the conditions are overall moist, revealing RH generally above 90 % (Phase 2, *Moist morning phase*, 3–8 UTC). After sunrise (6 UTC) the RH decreases again due to temperature increase and lifting of the stratus layer. Until 15 UTC the AI front re-establishes. This time period is denoted as Phase 3 (*Daytime drying phase*, 9–15 UTC). The comparison of Figure 3a and Figure 3b underlines that RH governs the spatiotemporal pattern of ALWC. Highest ALWC

values are reached in Phase 2 and especially in the hilly terrain north of 7.5° N (Fig. 3b). The AI front denotes a clear border of a non-negligible ALWC regime post-frontally and negligible ALWC pre-frontally. The study of Bian et al. (2014) found average ALWC values of about 170 g m$^{-3}$ for the North China plain, which is on the same order of magnitude as presented in Figure 3b.

In Phase 1 a cloud band develops behind the front which intensifies north of 7.5° N due to orographic lifting as visible in the

high RH in the Hovmöller diagram after 0 UTC. After 21 UTC further clouds, originating from the Gulf of Guinea, propagate inland. This is reflected in the high RH after 21 UTC south of approximately 6.5° N. Figure 3 reveals the strong impact of atmospheric dynamics, in particular AI, on the spatiotemporal evolution of RH. This is most pronounced in Phase 1. Without AI and the land-sea contrast, a zonally more homogeneous pattern would be expected for the diurnal cycle. In fact, this can only be observed in Phase 2. In this time period the zonal differences that developed during daytime have been removed by

the progressing AI. However, also the nighttime radiative cooling contributes to the increase in RH in addition to the cold air advection. When considering 6–7 July 2016 (see Fig. C1 in Appendix C), Phase 2 appears to be moister and the area south of the coast is drier but nevertheless the general evolution of RH and the three phases agree to the findings obtained for 2–3 July 2016, including the double-peak structure in Phase 2 with one peak near the coast and one peak in the hilly terrain to the north. The subsequent sections assess whether distinct differences in the ALWC and its impact on radiation can be identified between

the proposed phases.



### 3.3 Impact of aerosol modes

Figure D1 in Appendix D shows spatiotemporal mean vertical profiles of aerosol mass concentrations over Ivory Coast. Highest contributions of about three quarters come from organic aerosol as the sum of primary organics (POA) and secondary organics (SOA). The spatiotemporal mean reveals aerosol profiles that are rather constant with height in the lowest 2000 m AGL, only

for organic aerosol an increase with height is visible.

Figure 4 shows the relative ALWC (Fig. 4a-d) and (absolute) ALWC (Fig. 4e-h) for the different aerosol modes. The relative ALWC is related to the water mass absorbed by 1 $cm^{-3}$ of dry aerosol and the absolute ALWC to the water mass in an air volume of 1 $m^{-3}$. As described above, the ALWC dominates in the post-frontal area and especially in Phase 2 (Fig. 4e). This pattern is also visible for the relative ALWC (Fig. 4a). The main contribution comes from ACC (Fig. 4g). With respect to the

relative ALWC, COARSE shows highest water uptake per unit volume (Fig. 4d) south of approximately 8° N. The peak in the relative ALWC of AIT in Phase 2 (Fig.4b) might be related to high aerosol concentrations coming from the east. When focusing on the relative ALWC, AIT particles show a higher water uptake per unit volume than ACC particles (compare Fig. 4b and Fig. 4c). Nevertheless, due to their small size, ACC particles contribute the largest absolute ALWC (compare Fig. 4f and Fig. 4g). In fact, ACC is dominating in the ALWC contribution, because a sufficient number is available and the particles are

not too small. In contrast, AIT particles are lacking in size and COARSE particles in number. Accumulation mode particles over the Gulf of Guinea emerging to a large extent from long-range transport of biomass burning aerosols from central Africa (e.g. Mari et al., 2008) but there are also contributions from shipping emissions. Therefore it is expected that these particles are comparably old and therefore highly hygroscopic. Over land, emissions from cities contribute to the total aerosol amount. The spatial mean diurnal cycle of the particle number concentration reveals a decrease from about 5500 on 2 July 15 UTC to 4000

on 3 July 7 UTC with the inland advection of postfrontal air coming from the Gulf of Guinea. After sunrise the particle number concentration increases again with the evolution of the convective PBL (not shown). Figure 5 shows the mean GF (Fig. 5a) and the GF of the single aerosol modes (Fig. 5b-d). On average, GF of about 3–3.5 can be found in the high-ALWC areas of Phase 2 (Fig. 5a). The GF for AIT and ACC (Fig. 5a,b) are similar around 2.5, which is on the same order of magnitude compared to the findings of Chen et al. (2012) indicating GF of 1–3.25 for the North China Plain. As expected, highest aerosol

growth due to ALWC is observed for COARSE with values up to 4.5 (Fig. 5d). Sea salt particles are initially large and also highly hygroscopic but the number density is low (not shown). The spatial median of the GF is shown in Figure 6 revealing that AIT and ACC particles can be assumed comparably dry during Phase 1. During Phase 2 a doubling of the size can be observed. The comparably dry COARSE particles, already twice as big in Phase 1, double their diameter again within Phase 2. The absolute values of aerosol dry and wet diameters on 3 July 6 UTC (denoting the maximum GF in Fig. 6) are presented in

Figure E1 of Appendix E, indicating a substantial aerosol increase with water uptake. However, except of the largest sea salt mode (COARSE$_3$ in Fig. E1), which is related to very low number concentrations, all aerosol particles are below the typical size of a cloud droplet, which is on the order of magnitude of 10 $\mu$m. As expected, the highest GF can be found around sunrise related to the lowest temperatures and highest RH. When focusing on ACC and COARSE particles, the slope before 6 UTC is flatter than after 6 UTC, indicating that the GF enhancement from AI-induced RH increase is slower than the heating-induced





RH-decrease after sunrise. This is likely due to the combination of near-surface heating and lifting of the moist layer to greater heights. The aerosol growth, quantified in Figure 5 and 6, suggest substantial effects on the radiative transfer. This will be assessed in Section 4.

### 3.4 Impact of clouds

A further aspect that might affect the ALWC are clouds, as a special case of the RH depencency described in Section 3.2, with a focus on regimes that are saturated or virtually saturated with water vapor. It is an open question how much the ALWC in cloudy areas contribute to the total ALWC. In this section the total vertical column is considered. Figure 7 shows the total water column with respect to ALWC (Fig. 7). The respective value of the clouds is added to allow a comparison. As identified in Figure 3b, largest ALWC values are reached in Phase 2. However, also Phase 3 shows remarkable contributions that are

not visible when focusing on the layer below 1500 m AGL. The median cloud water is about 2–3 orders of magnitude larger (grey curve in Fig. 7), but due to the large standard deviation, ALWC and cloud water can differ 4 orders of magnitude (10th percentile). ALWC and cloud water correlate in the diurnal evolution (Fig. 7) with one peak in Phase 2 (NLLS) and one peak in Phase 3 (convective clouds). Interestingly, the cloud water and also the ALWC show a local minimum in the transition between Phase 2 and Phase 3 during the SCT.

Figure 8 quantifies the contribution of ALWC that comes from cloudy grid volumes. The in-cloud contribution from $\mathrm{ALWC_{Total}}$ is between 40 and 60 %, clearly dominated by $\mathrm{ALWC_{AIT}}$ and $\mathrm{ALWC_{ACC}}$. The in-cloud contribution of $\mathrm{ALWC_{COARSE}}$ is smaller with a constant diurnal offset of about 20 % compared to the other aerosol modes. Although the strong contribution of in-cloud areas to the ALWC is not surprising, since here the highest RH can be expected, it is nevertheless remarkable. Between 3 and 9 % of the model grid boxes in the lowest 10 km are related to clouds during the day and this small fraction captures more

than half of the total ALWC. To highlight the importance of this finding, the contribution of in-cloud AOD to the total AOD is added in Figure 8, which shows the same diurnal evolution as the ALWC. Approximately 40 % of the total AOD is related to cloud areas.

## 4   ALWC impact on radiative transfer

### 4.1   Definition of subdomains

After assessing the quantities affecting the ALWC (Sect. 3), this section focuses on the impact ALWC has on the radiative transfer comparing *No-ALWC* with *Reference*. *No-ALWC* denotes a sensitivity study neglecting the ALWC in the radiative transfer calculations. To evaluate the differences in net downward shortwave radiation at the surface (SSR) and net downward longwave radiation at the surface (SLR) between the two realizations, it is necessary to consider side-effects that have the potential to affect the differences apart from the consideration of the ALWC, in particular spatiotemporal differences in cloud

pattern (displacement of clouds). Therefore, in the following two domain subsets are considered: (1) areas that are simultaneously cloudy in both realizations (in-cloud area, ICA) and (2) areas that are simultaneously cloud free in both realizations





(off-cloud area, OCA). Areas, which differ in the cloud status, are omitted. Even if a grid box is related to clouds in both realizations, cloud properties may differ. Statistics of the difference in cloud properties are summarized in Table F1 of Appendix F including the full time period 2 July 15 UTC to 3 July 15 UTC. Table F1 includes the total cloud water, Cloud Droplet Number Concentration (CDNC) and effective radius. The spatiotemporal median over Ivory coast reveals negligible differences in

cloud properties. However, for spatial analyses substantial differences can occur due to a displacement of clouds and different properties. Therefore it is not possible to fully disentangle radiative effects of ALWC from the cloud displacement in ICA. This is especially problematic since ICA is related to the highest ALWC amounts as shown in Section 3.4. In the following, we sharpen the condition for ICA by considering only the areas in which the total cloud water differences between the two realizations are below 0.1 g m$^{-2}$ (approximately 1 % of the *Reference* median). Consider that the sharpened condition substantially

decreases the selected area and therefore makes the results less representative for the cloudy area. OCA is expected to provide more robust results since the properties of clouds are not relevant in this area.

### 4.2 Spatiotemporal differences in near-surface atmospheric properties

Figure 9 shows SSR in terms of the *Reference* absolute values (Fig. 9a) and the difference between *Reference* and *No-ALWC* (Fig. 9b) as a Hovmöller diagram. The following values in brackets indicate the median and the $99^{th}/1^{th}$ percentile of the

differences considering the area south of 8° N. Since Phase 1 and 2 are related to the evening and night, the ALWC-SSR impact is restricted to Phase 3 and the early hours of Phase 1. When focusing on the area south of 8° N generally a decrease in SSR can be observed when considering ALWC in the radiation for ICA (-28 W m$^{-2}$, -111 W m$^{-2}$; Fig. 9b) and OCA (-15 W m$^{-2}$, -107 W m$^{-2}$; Fig. 9c). The positive values north of 8° N in Phase 3 are related to a change in cloud cover, which is not a general feature. On 6-7 July the entire domain is related to a reduction in SSR (Fig. G2). In Figure 10 the SLR is shown with

respect to the *Reference* absolute value (Fig. 10a) and the difference between *Reference* and *No-ALWC* (Fig. 10b). For SLR the differences are less coherent than for SSR and the values are much smaller. Areas with positive and negative differences occur. For the post-frontal area, especially in the late Phase 1 and in Phase 2, negative values prevail, which indicates more outgoing longwave radiation in *No-ALWC*. Without the ALWC, less SLR can be absorbed and re-emitted in the atmosphere. During Phase 3 SLR is reduced due to the reduced shortwave input (compare Fig. 9b,c). The SLR differences are small in ICA

(-0.6 W m$^{-2}$, -10 W m$^{-2}$; Fig. 10b) and OCA (-0.5 W m$^{-2}$, -13 W m$^{-2}$; Fig. 10c) in agreement with the findings on 6–7 July (Fig. G2). Especially the nighttime SLR differences appear insignificant. Only during daytime, with changes in SSR, the SLR shows relevant differences when considering the ALWC. The radiative impact on 2-m temperature is presented in Figure 11a. When focusing on the absolute values (Fig. 11), a diurnal cycle of about 8 K can be observed inland. Due to the fixed Sea Surface Temperature (SST) in COSMO-ART, the air temperature over the Gulf of Guinea is virtually constant. The definition

of the three phases (Fig. 3) agrees well to the diurnal cycle of the temperature. In Phase 2 lowest temperatures over entire Ivory Coast can be observed. Generally, the daytime heating inland (pre-frontal in Phase 1 and 3) is stronger in *No-ALWC* than in *Reference* due to additional SSR input (Fig. 9b,c).

As expected from the small nighttime difference in SLR, also no relevant temperature differences occur during night (Fig. 11b,c). The postfrontal area (Phase 1 and 2), which is related to airmasses from the ocean with fixed SST, eliminates the differences



developing during day. The temperature difference during Phase 1 and 3 are negative for ICA (-0.04 K, -1.2 K; Fig. 11b) and OCA (-0.04 K, -1.3 K; Fig. 11c). The differences in SSR, SLR and 2-m temperature on 6–7 July, which are given in Figure G1, G2 and G3 of Appendix G, are on the same order of magnitude as on 2–3 July. Also the comparison between the results for ICA and OCA reveals no significant differences.

## 4.3 ALWC impact on Aerosol Optical Depth (AOD)

The observed changes in radiative transfer due to ALWC are caused by the ALWC impact on the AOD. Figure 12 presents the Empirical Cumulative Distribution function (ECDF) for the modeled AOD using the entire 25 h time period separated to ICA (blue) and OCA (red) AOD. With this the authors want to highlight that the focus is on aerosols and the AOD and not on effects of the cloud optical thickness. When neglecting ALWC in the radiation, the AOD distribution is virtually equal for ICA and OCA (dashed lines in Fig. 12) with median values around 0.2 (circles). In contrast, the AOD distribution significantly differs when considering ALWC (solid lines in Fig. 12) with median values of 0.7 (ICA, blue dot) and 0.3 (OCA, red dot). We conclude that ALWC generally increases the AOD and also causes AOD sensitivities with respect to RH (ICA and OCA). Despite the substantial differences in AOD between the two realizations with respect to ICA, the differences in SSR, SLR and 2-m-temperature are not significantly higher than for OCA in the zonal mean as shown above. Most likely clouds are dominating the radiative transfer in ICA and therefore the AOD has less impact in these areas. The diurnal cycle of the AOD is shown in Figure 13 for *Reference* and *No-ALWC*. Without ALWC, the AOD is rather zonally-symmetrical without a remarkable diurnal evolution (Fig. 13b) but by including ALWC, a clear diurnal cycle emerges (Fig. 13a). The dry areas, in particular the pre-frontal area in Phase 1, show AOD minima, whereas the wetter Phases 2 and 3 reveal a significant AOD increase.

Figure 14 summarizes the ALWC effects on SSR, SLR and AOD by presenting the differences between *Reference* and *No-ALWC* for the entire Ivory Coast. Strongest signals are visible for SSR during Phase 1 and 3. Differences in the SLR are likely related to cloud fraction variations (note anticorrelation of SSR and cloud fraction differences). The fluctuations in SSR (red solid line) after sunrise are related to differences in the cloud cover (blue line). The AOD is higher in the wet Phases 2 and 3 than during the comparably dry Phase 1.

## 4.4 ALWC-radiation relationship

Sections 4.2 and 4.3 provided insight in the ALWC effects on radiation and AOD. Based on these findings the pivotal question is: Can we observe a robust relationship between ALWC and SSR as well as SLR? To answer this question we used the full 25 h period 2 July 15 UTC to 3 July 15 UTC and clustered it according to clouds (ICA and OCA) and time of day (daytime: 2 July 15–18 UTC and 3 July 7–15 UTC; nighttime: 2 July 19 UTC to 3 July 6 UTC). The clusters are sorted according to the total column ALWC (bin size 0.01 g m$^{-2}$). For all grid points, which are assigned to a certain bin, the median of the radiation differences *Reference* minus *No-ALWC* is calculated and plotted (blue lines in Fig. 15) together with the 25$^{th}$ and 75$^{th}$ percentile (blue shading). Linear fits are added to an empirically selected subset of the total ALWC range, omitting the bins with large ALWC (less data and large spread) and low ALWC due to the nonlinearity, to quantify the ALWC-radiation relationship. The





slopes (W g$^{-1}$), which are derived from the linear fitting, are summarized in Table 1. Furthermore, we applied a bootstrapping technique for the six ALWC-radiation datasets of Table 1. For 10.000 re-samples the corresponding slopes are calculated to estimate the uncertainty of the slope (Table 1). Nevertheless, the informative value of this approach is limited to the fact that the ALWC-radiation relationship is not only defined by the ALWC itself but also by the distribution of the ALWC on aerosol

particles. It can be expected that with the same total ALWC, many small particles with small ALWC values are more effective in altering the radiation than a few big particles with high ALWC values. Therefore, it might by problematic to compare these results with other regions with different aerosol distributions. Generally, the increase in ALWC leads to a decrease in SSR and SLR in *Reference* compared to *No-ALWC* (Fig. 15), which is more pronounced for ICA (Fig. 15, left) than for OCA (Fig. 15, right). Since ICA covers a wider ALWC interval than OCA, also the linear fit is more robust. In Section 4.1 ICA is defined

as an area which is affected by clouds in both realizations *and* the differences in total cloud water are below 0.1 g m$^{-2}$ to minimize effects from displaced clouds. This cloud water threshold value is generally smaller than the observed values of ALWC suggesting that effects from cloud water differences are smaller than effects from ALWC. Highest ALWC-radiation sensitivities can be observed for SSR in ICA with about -300 W g$^{-1}$. For OCA the decrease is about -100 W g$^{-1}$. With respect to SLR, a separation in daytime and nighttime is done, with the former referring to the time period used for the SSR analysis.

Negative SLR differences denote more outgoing longwave radation in *No-ALWC*. This indicates that the ALWC contributes to the absorption and re-emission of SLR in the atmosphere. What we learn from the bootstrapping is that there are no statistically significant differences between the SLR decrease ICA and OCA or during daytime and nighttime (Table 1). The decrease is on an order of magnitude of -10 W g$^{-1}$. Interestingly, for ALWC values below 0.05 g m$^{-2}$ positive differences occur (more outgoing longwave radiation in *Reference*) and the relationship is nonlinear (Fig. 15c-f, but also 15a). Therefore the linear

fit omits this part of the curve. The reason for this behavior is not clear. This analysis is repeated for 6–7 July (Fig. H1 and Table H1, confirming the general relationship of decreasing radiation with increasing ALWC values. For SSR ICA, similar results are found whereas the other subsets tend to have stronger radiation declines with ALWC than on 2–3 July 2016. On the one hand this is related to the sensitivity of the interval selection for the fitting and on the other hand 6–7 July shows higher RH (Fig. C1a) and lower temperatures (Fig. G3a) than 2–3 July and therefore a higher potential for altering the radiation.

**5  Conclusions**

This modeling study set the focus on the impact of Aerosol Liquid Water Content (ALWC) on the radiative transfer over Southern West Africa (SWA). It provides a complementary study to Deetz et al. (2018), which focuses on the implication of aerosols on clouds and the atmospheric dynamics over SWA. The results are obtained via a process study with the regional model COSMO-ART on 2-3 and 6-7 July 2016, a time period in the well-established West African Monsoon (WAM) and

little impacts of Mesoscale Convective Systems. With our study we aimed at (1) the quantification of the diurnal evolution of ALWC-related properties, (2) the evaluation of the ALWC impact on radiative transfer and (3) to derive robust relationships between ALWC and the change in radiative transfer.

Deetz et al. (2018) identify the Atlantic Inflow (AI) as an atmospheric phenomenon, which affects entire SWA by changes in





temperature, relative humidity (RH) and wind speed with an especially coherent pattern over Ivory Coast. Therefore the spatial focus in Deetz et al. (2018) and this analysis is on this area. It turns out that AI, as an underlying meteorological process, affects all measures relevant for ALWC, in particular RH, clouds and aerosol properties. AI affected the monsoon layer (lowest 1900 m AGL) by advecting airmasses with comparably low temperatures and high RH onshore. Highest RH are reached in the

post-frontal area of AI. We have shown that AI decisively shapes the diurnal evolution of the RH and propose three phases: *Phase 1* (15–2 UTC) denotes the progression of the AI, inducing an inland contrast between the comparably dry and warm air pre-frontally and the comparably moist and cold are post-frontally. *Phase 2* (3–8 UTC) refers to the moist morning. The AI front has passed the area providing homogeneously moist and cool conditions. *Phase 3* (9–15 UTC) is the Daytime drying phase. After sunrise the land area warms and dries again leading to the re-establishment of the AI front. Due to AI the diurnal

cycle is not primarily thermodynamically driven (nighttime radiative cooling) but dynamically driven. Since several studies (e.g. Adler et al., 2017) have shown that AI is a common phenomenon during the West African monsoon (WAM), we suggest that the proposed phase definition can be generalized to this time period. This is supported by additional simulations for 6–7 July that show similar patterns. The spatiotemporal pattern of ALWC is clearly dominated by that of the RH. On average 60–80 % of the ALWC is related to RH regimes > 95 %. With respect to the aerosol size, the accumulation mode is the dominant

ALWC contributor in agreement to the findings of Bian et al. (2014). These particles are adequate in number and size and are also highly hygroscopic. Around sunrise (6 UTC, Phase 2) highest RH and therefore the ALWC maximum is reached over SWA. This is related to aerosol growth factors of about 2 for Aitken and accumulation mode and about 4 for coarse mode particles.

The radiative impact of ALWC is assessed by an additional model realization that neglects the ALWC impact on the radiative

transfer. Including the ALWC leads to a significant increase in the Aerosol Optical Depth (AOD) especially for cloud areas (from about 0.2 to 0.7 on average). Therefore ALWC introduces a RH dependency in the AOD. However, effects from the AOD increase in cloudy areas on shortwave radiation and temperature are not significantly stronger than for the areas off clouds, likely because the clouds are dominating the radiative transfer and the AOD has less impact. Generally, a decrease in incoming shortwave radiation can be observed when considering ALWC on an order of magnitude of -20 W m$^{-2}$ (spatiotemporal

average). Longwave effects appear insignificant. Since the effects are small during night, also 2-m temperature differences are restricted to daytime. The temperature decrease is usually not greater than -1 K but this is already significant in moist tropical climates.

To derive a relationship between ALWC and radiation (W per g ALWC), we calculated linear fits to the radiation decrease with increasing total column ALWC and estimated the uncertainty by using a bootstrapping technique. For shortwave radiation

in-cloud (off-cloud), a relationship of -305±39 W g$^{-1}$ (-114±42 W g$^{-1}$) is found. For longwave radiation the relationship is about -10 W g$^{-1}$ with insignificant differences between day and night as well as in-cloud and off-cloud. However, these relationships do not include effects arising from the aerosol optical properties (many small particles versus few large particles). The findings indicate the general need to consider ALWC or the RH dependency of the AOD in the COSMO radiation calculation. This is especially of relevance in SWA with its moist and polluted monsoon layer. Although, the additional period 6–7 July

is used to evaluate the robustness of the results, revealing similar evolutions of AI and the radiation differences, simulations




on longer time scales are necessary to increase the reliability in the ALWC-radiation relationship. A drawback in this study is that the activated aerosol is not removed from the aerosol distribution leading to potential double counts in the radiative transfer calculations. A simulation with radiatively fully transparent clouds can provide further insight in the ALWC-radiative impact by disentangling from the cloud properties but it is expected that the surplus in incoming solar radiation would sig-

nificantly alter the atmospheric dynamics and therefore make it less realistic. The non-negligible radiative impact of ALWC motivates post-DACCIWA measurement efforts in which the SWA haze could be targeted. In this regard the time of sunrise will be of special interest, since at this time the ALWC maximum is reached and also the humidity related AOD increase is highest. However, strongest effects on temperature occurs later in the morning. The quantification of aerosol hygroscopicity with aircrafts on clear and hazy days might allow to derive observational-based relationships between ALWC and the radiative

transfer or visibility in general. Especially nocturnal research flights can provide added value complementary to DACCIWA. An interesting time of year to further study this effect is boreal spring (e.g. March) characterized by pre-monsoon conditions with high aerosol and humidity but less cloud and precipitation than in summer. There will be a measurement study estimating the ALWC by using aircraft observations and the ZSR mixing rule (S. Haslett, personal communication).

*Data availability.* The underlying research data are available upon request from the corresponding author.

*Competing interests.* The authors declare that they have no conflict of interest.

*Special issue statement.* This article is part of the special issue *Results of the project "Dynamics–aerosol–chemistry–cloud interactions in West Africa" (DACCIWA)*



List of acronyms used in this study.

| Acronym | Description |
| --- | --- |
| ACC | Accumulation mode |
| ADE | Aerosol Direct Effect |
| AGL | Above Ground Layer |
| AI | Atlantic Inflow |
| AIE | Aerosol Indirect Effect |
| AIT | Aitken mode |
| ALWC | Aerosol Liquid Water Content |
| AOD | Aerosol Optical Depth |
| ASL | Above Sea Level |
| CDNC | Cloud Droplet Number Concentration |
| COARSE | Coarse mode |
| COSMO-ART | Consortium for Small-scale Modeling - Aerosol and Reactive Trace gases |
| DACCIWA | Dynamics-aerosol-chemistry-cloud interactions in West Africa |
| DWD | Deutscher Wetterdienst (German Weather Service) |
| ECDF | Empirical Cumulative Distribution Function |
| GF | Growth Factor |
| GRAALS | General Radiative Algorithm Adapted to Linear-type Solutions radiation scheme |
| HaChi | Haze in China campaign |
| ICA | In-Cloud Area |
| ICON | Icosahedral Nonhydrostatic Model |
| No-ALWC | Model realization neglecting ALWC in the radiation calculation |
| OCA | Off-Cloud Area |
| PBL | Planetary Boundary Layer |
| POA | Primary Organic Aerosol |
| Reference | Reference case model realization with considering ALWC in the radiation calculation |
| RH | Relative Humidity |
| SCT | Stratus-to-cumulus transition |
| SLR | Surface Longwave (net) Radiation |
| SOA | Secondary Organic Aerosol |
| SSR | Surface Shortwave (net) Radiation |
| SST | Sea Surface Temperature |
| SWA | Southern West Africa |
| WAM | West African Monsoon |





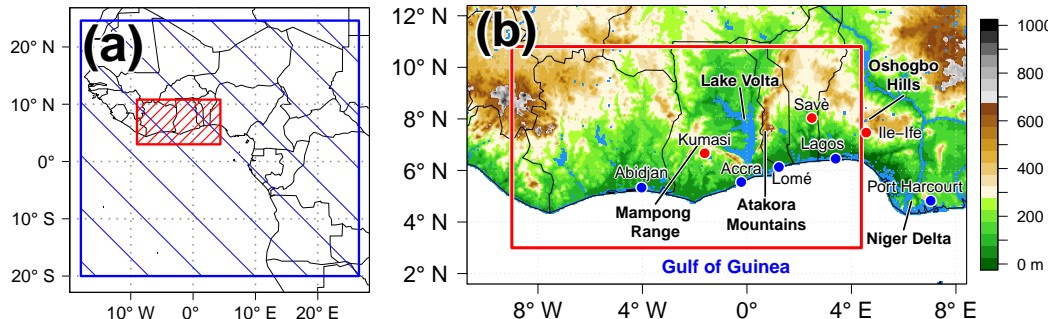

**Figure 1.** (a) Modeling domain SWA (red rectangle, 2.5 km grid mesh size) together with its coarse domain (blue, 5 km grid mesh size). (b) Map of the research area SWA. The color shading denotes topography (m Above Sea Level, ASL). Topographic features are named in bold, coastal cities are shown as blue dots and the three DACCIWA supersites as red dots. The modeling domain SWA is again denoted as red rectangle. Figure adopted from Deetz et al. (2018).

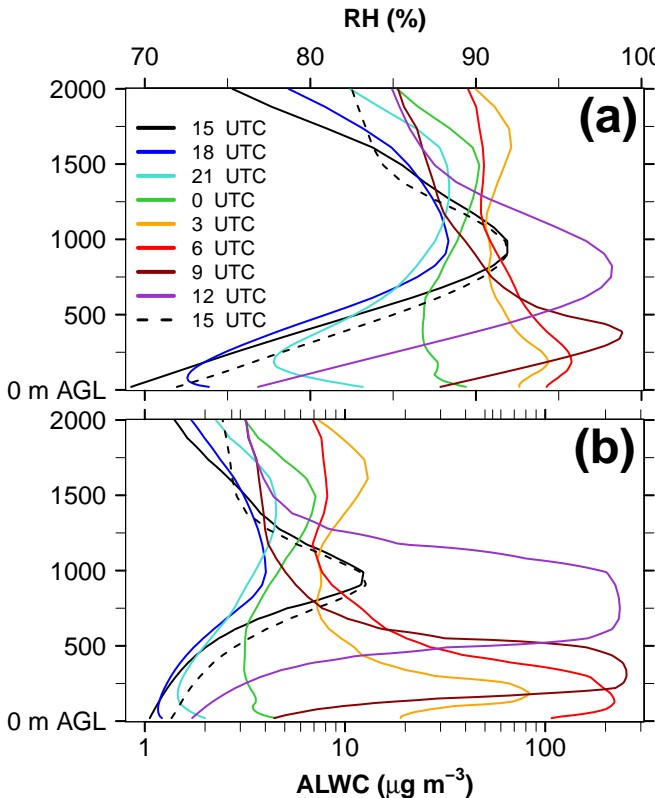

**Figure 2.** Vertical profiles (m AGL) of (a) RH (%) and (b) ALWC ($\mu$g m$^{-3}$) for the median over Ivory Coast (7.5° W–3° W, 4–10° N) between 2 July 15 UTC (black solid) and 3 July 15 UTC (black dashed). Consider the logarithmic abscissa of (b).



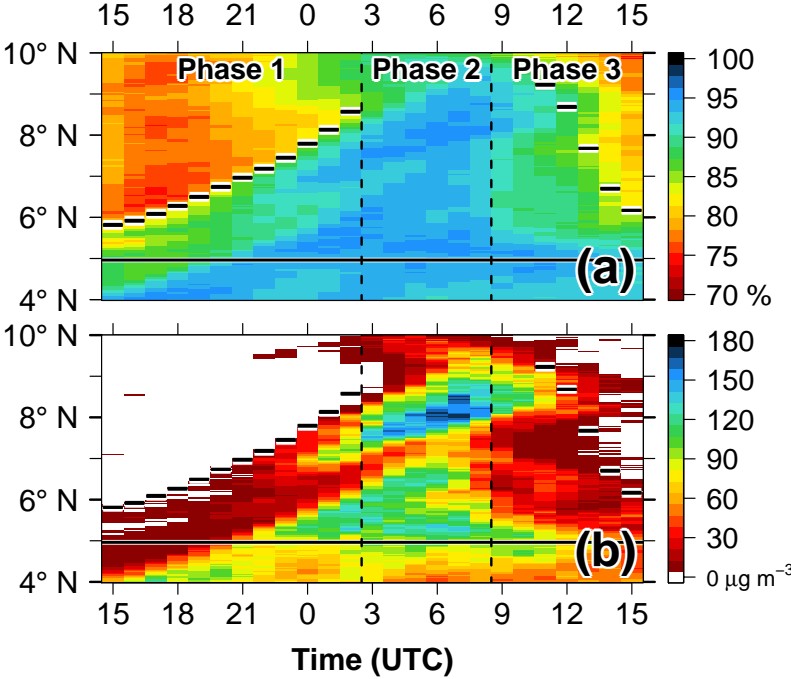

**Figure 3.** Hovmöller diagram of the median (a) RH (%) and (b) total ALWC ($\mu$g m$^{-3}$) in the lowest 1500 m AGL as zonal mean over Ivory Coast (7.5° W–3° W, 4–10° N) between 2 July 15 UTC and 3 July 15 UTC. The horizontal bars denote the zonal mean location of the 302 K isentrope at 250 m AGL, the horizontal solid line the zonal mean coast line and the vertical dashed lines separate the three phases: *AI progression phase* (Phase 1), *Moist morning phase* (Phase 2) and *Daytime drying phase* (Phase 3).

**Table 1.** Radiation-ALWC relationship (W g$^{-1}$) based on linear fits as presented in Figure 15, including the time period 2 July 15 UTC to 3 July 15 UTC. The subdomain denotes whether the captured area is simultaneously cloudy (ICA) or cloud-free (OCA) in both realizations. *Original data* denote the slopes derived from Fig. 15 whereas *bootstrapping* refers to the median slopes of the 10.000 re-samples. The confidence intervals are derived by using the Gaussian approximation and $\alpha$=0.05 and the evaluation range provides the ALWC interval, which is used for the linear fitting.

| Radiation | Subdomain | Radiation-ALWC relationship (W g$^{-1}$) | | Evaluation range (g m$^{-2}$) |
|---|---|---|---|---|
| | | Original data | Bootstrapping | |
| SSR | ICA | -318 | -305±39 | 0.00–0.49 |
| SSR | OCA | -106 | -114±42 | 0.00–0.29 |
| Daytime SLR | ICA | -12 | -12±5 | 0.04–0.49 |
| Daytime SLR | OCA | -8.4 | -12±10 | 0.04–0.39 |
| Nighttime SLR | ICA | -7.1 | -7.1±1.3 | 0.05–0.50 |
| Nighttime SLR | OCA | -6.0 | -6.5±2.3 | 0.04–0.49 |





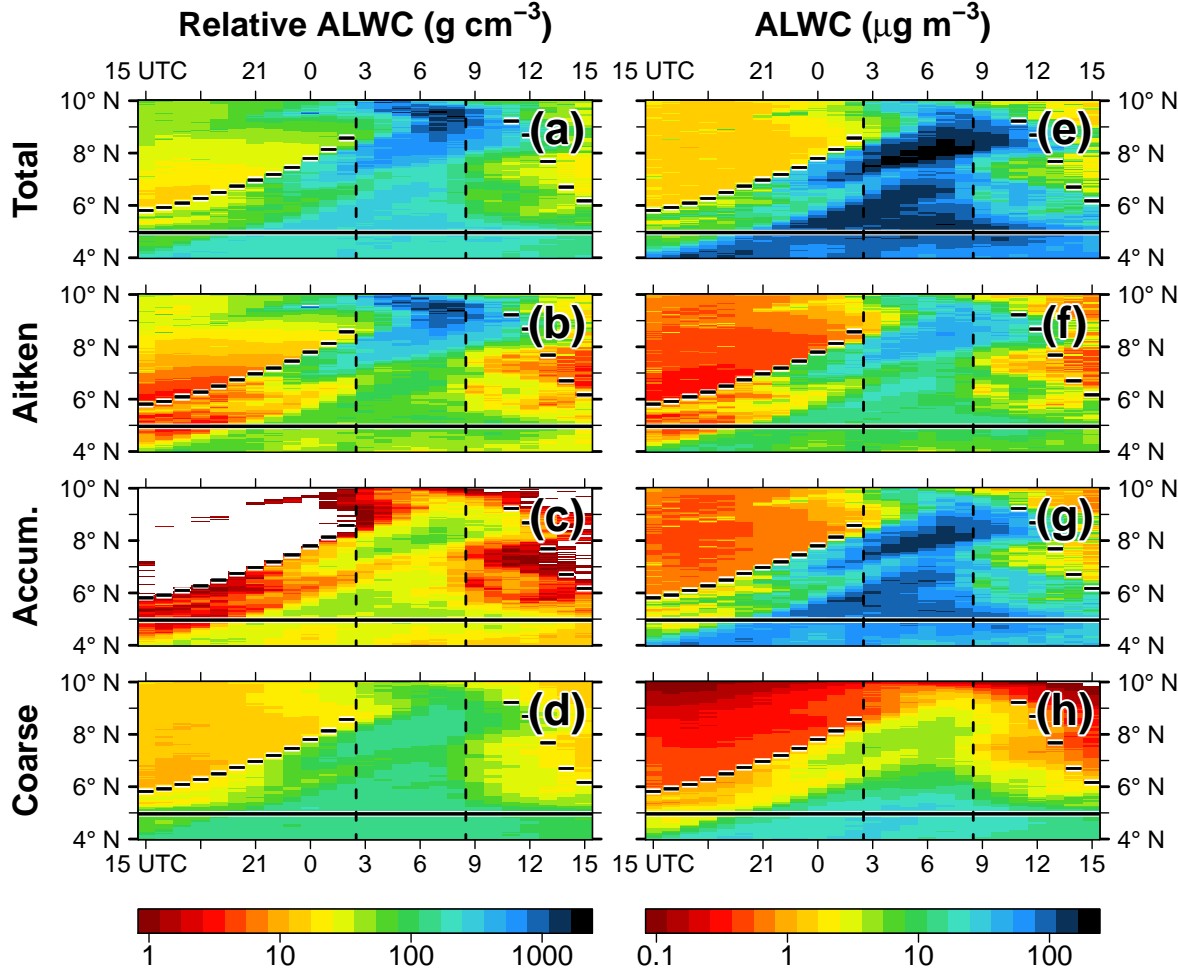

**Figure 4.** Same ass for Fig. 3 but for (left) relative ALWC (g ALWC cm$^{-3}$ dry aerosol) with respect to (a) total relative ALWC, (b) relative ALWC$_{AIT}$, (c) relative ALWC$_{ACC}$ and (d) relative ALWC$_{COARSE}$ and (right) absolute ALWC ($\mu$g ALWC m$^{-3}$ air) with respect to (e) total ALWC, (f) ALWC$_{AIT}$, (g) ALWC$_{ACC}$ and (h) ALWC$_{COARSE}$.





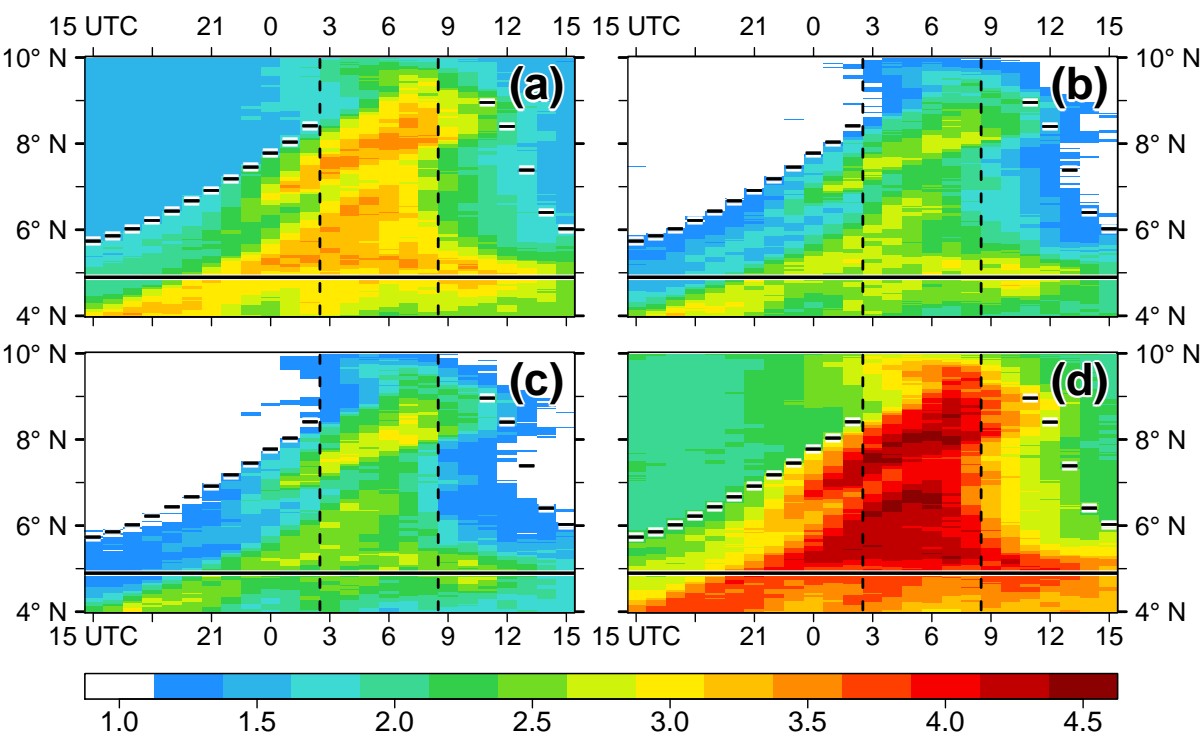

**Figure 5.** Same as for Fig. 3 but for the GF ($d_{p,\text{wet}}\, d_{p,\text{dry}}^{-1}$) with respect to (a) mean GF, (b) $\text{GF}_{\text{AIT}}$, (c) $\text{GF}_{\text{ACC}}$ and (d) $\text{GF}_{\text{COARSE}}$.





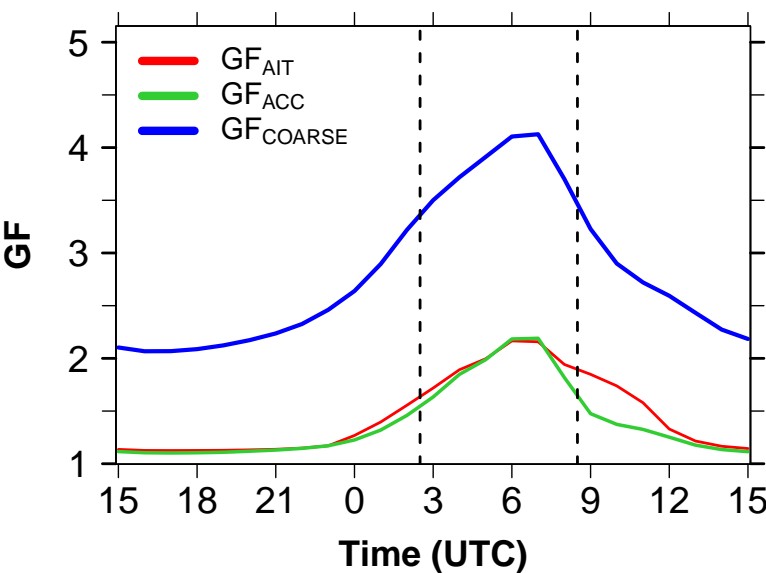

**Figure 6.** Diurnal cycle of the median GF (%) of $GF_{AIT}$ (red), $GF_{ACC}$ (green) and $GF_{COARSE}$ (blue) in the lowest 1500 m AGL over Ivory Coast (7.5° W–3° W, 4–10° N) from 2 July 15 UTC to 3 July 15 UTC. The vertical dashed lines denote the three phases introduced in Figure 3.

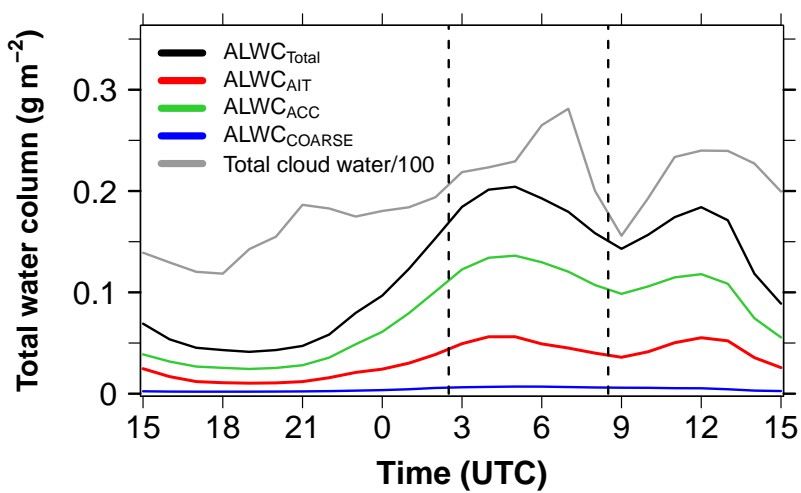

**Figure 7.** Diurnal cycle of total water column (g m$^{-2}$) as median over Ivory Coast (7.5° W–3° W, 4–10° N) from 2 July 15 UTC to 3 July 15 UTC with respect to $ALWC_{Total}$ (black), $ALWC_{AIT}$ (red), $ALWC_{ACC}$ (green) and $ALWC_{COARSE}$ (blue) as well as the median total cloud water (grey, divided by 100). Values below $10^{-3}$ g m$^{-2}$ are not considered. The vertical dashed lines denote the three phases introduced in Figure 3.




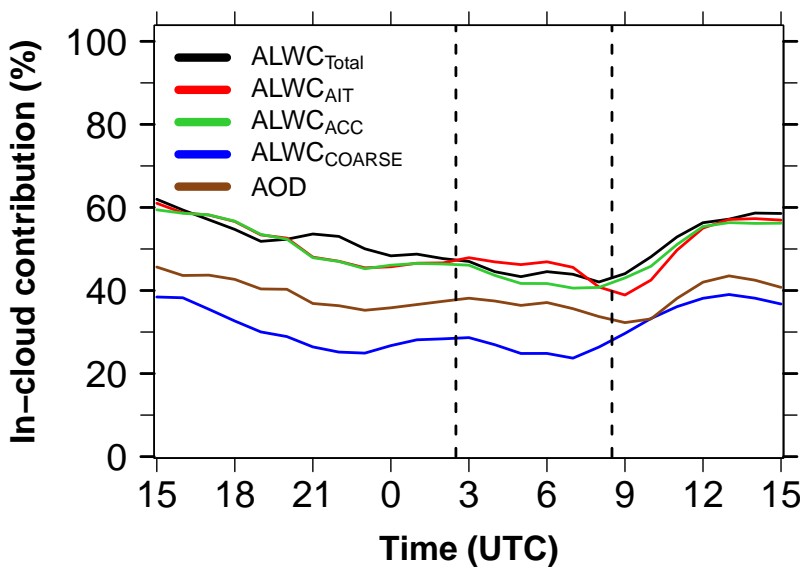

**Figure 8.** Diurnal cycle of the median contribution from in-cloud areas (%) with respect to ALWC$_{Total}$ (black), ALWC$_{AIT}$ (red), ALWC$_{ACC}$ (green), ALWC$_{COARSE}$ (blue) and the contribution of in-cloud AOD to the total AOD (brown) in the total vertical column over Ivory Coast (7.5° W–3° W, 4–10° N) from 2 July 15 UTC to 3 July 15 UTC. The vertical dashed lines denote the three phases introduced in Figure 3.




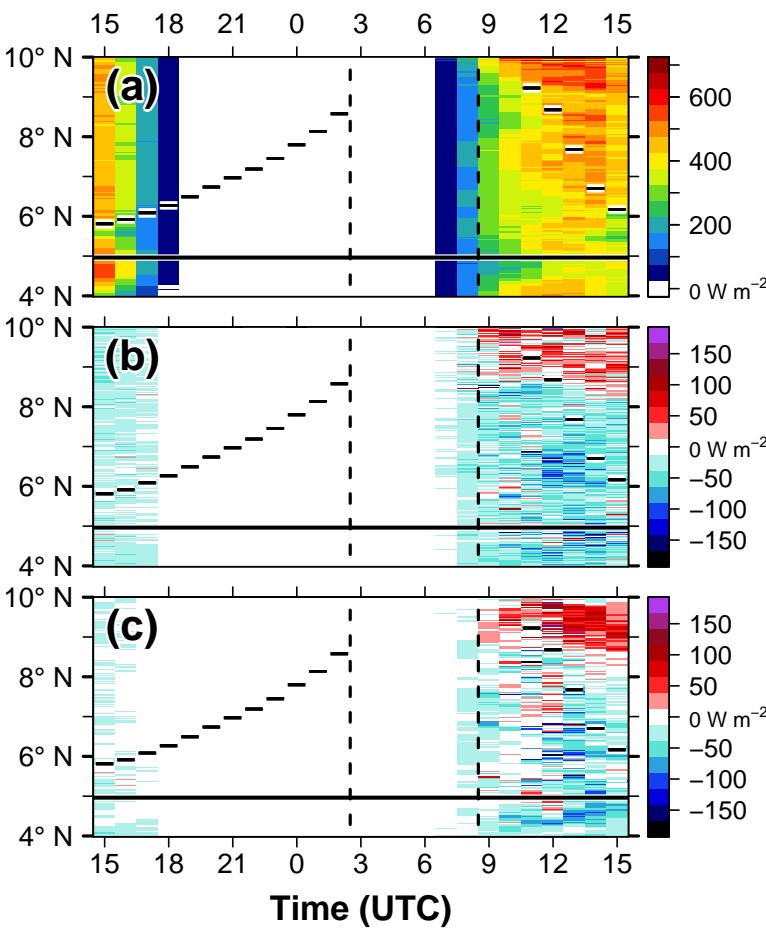

**Figure 9.** Hovmöller diagram of SSR (W m$^{-2}$) for (a) *Reference*, (b) *Reference* minus *No-ALWC* for ICA and (c) *Reference* minus *No-ALWC* for OCA as zonal mean over Ivory Coast (7.5° W–3° W, 4–10° N) between 2 July 15 UTC and 3 July 15 UTC. The horizontal bars denote the zonal mean location of the 302 K isentrope at 250 m AGL of *Reference*, the horizontal solid line the zonal mean coast line and the vertical dashed lines the three phases introduced in Figure 3.





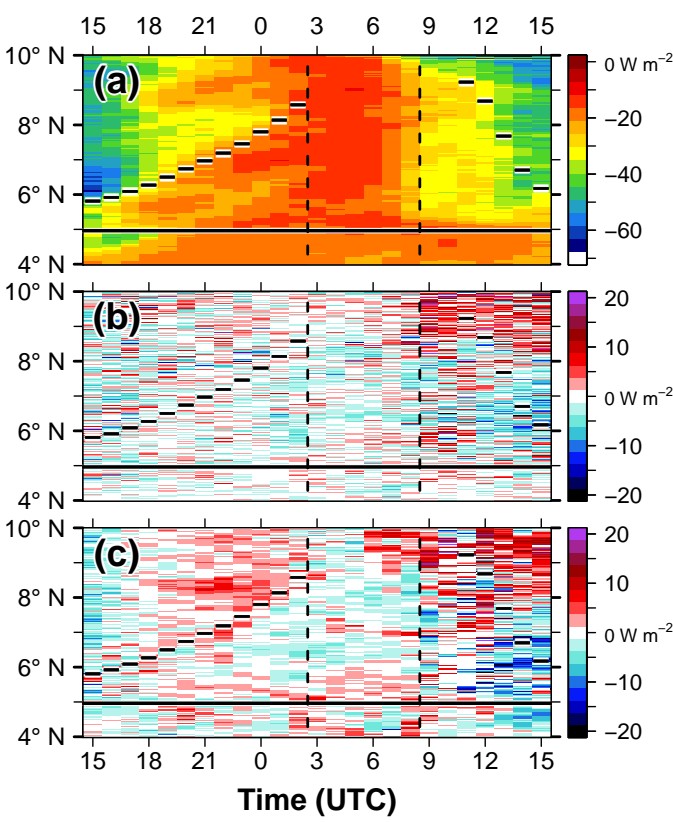

**Figure 10.** Same as for Fig. 9 but for SLR: (a) *Reference*, (b) *Reference* minus *No-ALWC* for ICA (c) *Reference* minus *No-ALWC* for OCA.
Positive (negative) values in (b) and (c) denote more outgoing longwave radiation in *Reference* (*No-ALWC*).



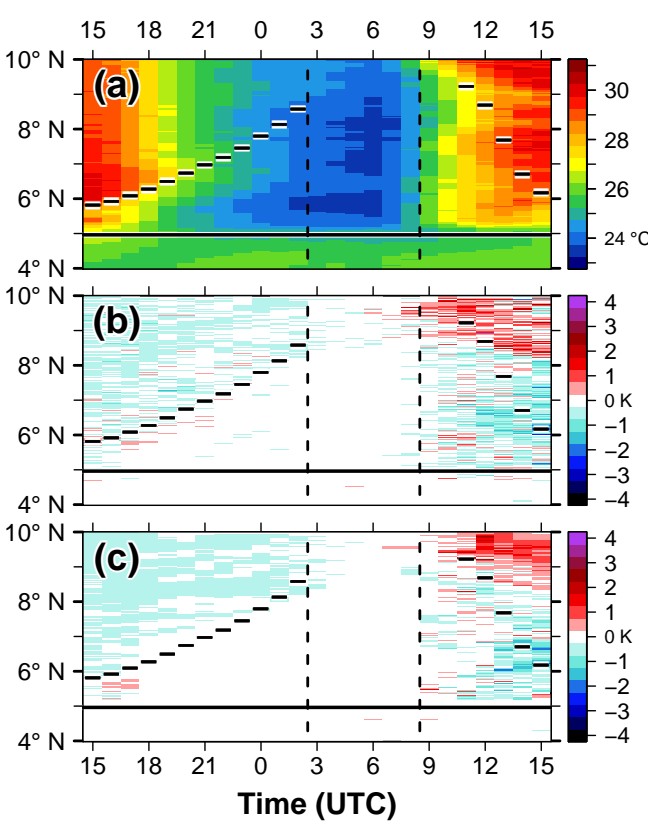

**Figure 11.** Same as for Fig. 9 but for 2-m temperature (°C) and 2-m temperature difference (K): (a) *Reference*, (b) *Reference* minus *No-ALWC* for ICA (c) *Reference* minus *No-ALWC* for OCA.





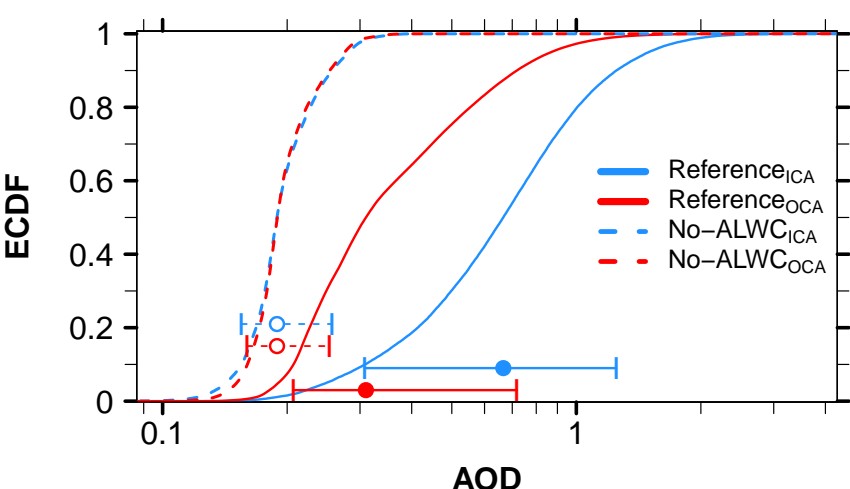

**Figure 12.** ECDF of the total AOD of ICA (blue) and OCA (red) for *Reference* (solid lines) and *No-ALWC* (dashed lines) over Ivory Coast (7.5° W–3°W, 4–10° N) including the time period from 2 July 15 UTC to 3 July 15 UTC. The dots (circles) highlight the median with respect to *Reference* (*No-ALWC*) and the whisker the 10th and 90th percentiles.




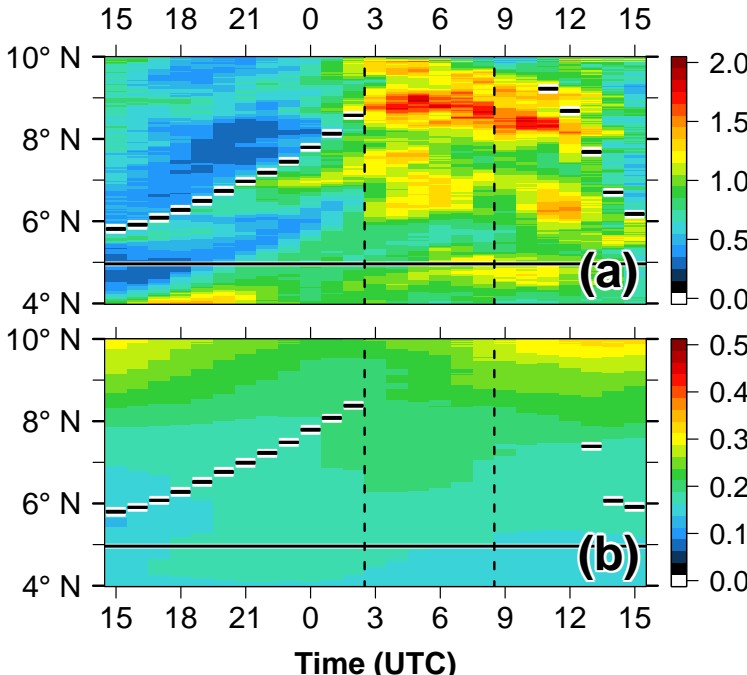

**Figure 13.** Hovmöller diagram of AOD for (a) *Reference* and (b) *No-ALWC* as zonal mean over Ivory Coast (7.5° W–3° W, 4–10° N) between 2 July 15 UTC and 3 July 15 UTC. The horizontal bars denote the zonal mean location of the 302 K isentrope at 250 m AGL, the horizontal solid line the zonal mean coast line and the vertical dashed lines the three phases introduced in Figure 3. Note the different color scales in (a) and (b).



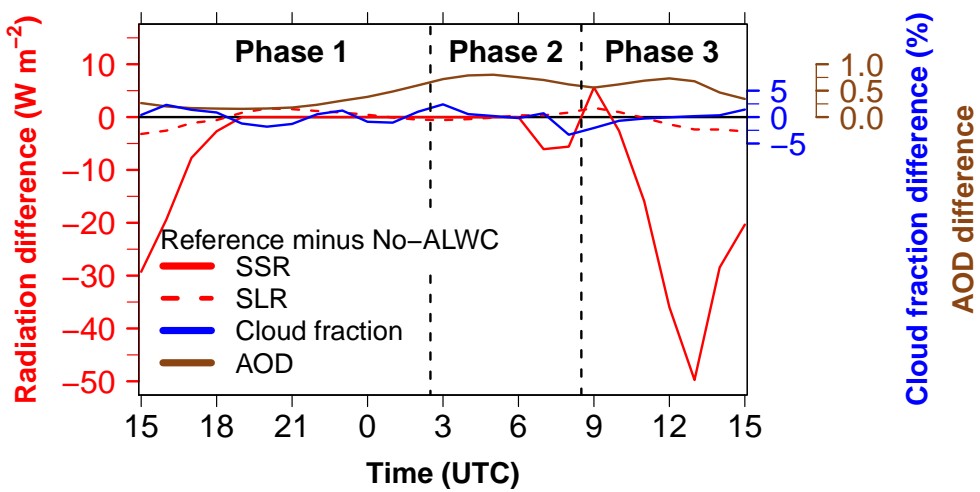

**Figure 14.** Diurnal cycle of the difference between *Reference* and *No-ALWC* with respect to SSR (red solid, W m$^{-2}$), SLR (red dashed, W m$^{-2}$), domain-wide cloud fraction (blue, %) and AOD (brown) as median over Ivory Coast (7.5° W–3° W, 4–10° N) between 2 July 15 UTC and 3 July 15 UTC. The vertical dashed lines indicate the three phases introduced in Figure 3. Note the color coded of the different ordinates.





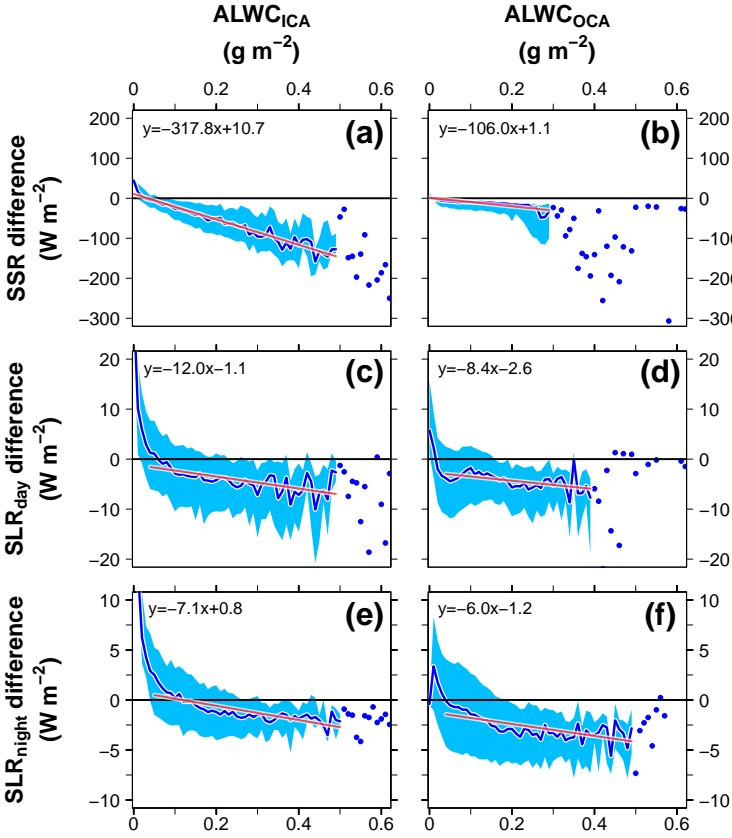

**Figure 15.** Relationship between the total column ALWC (g m$^{-2}$) and the radiation difference between *Reference* and *No-ALWC* (W m$^{-2}$) capturing Ivory Coast (7.5° W–3° W, 4–8° N) and the time period 2 July 15 UTC to 3 July 15 UTC. The data is clustered in areas that are simultaneously cloudy (left, ICA) or cloud free (right, OCA) in both realizations. The top panels show the SSR difference (2 July 15–18 UTC and 3 July 7–15 UTC), the middle panels the SLR daytime difference (same time period as SSR) and the bottom panels the SLR nighttime difference (2 July 19 UTC to 3 July 6 UTC). The ALWC values are clustered in bins with an increment of 0.01 g m$^{-2}$. For every bin the spatial median of the radiation difference is calculated (blue line). The envelope, spanned by the 25[th] and 75[th] percentile of the radiation difference, is shown as blue shading. For greater ALWC values the spread significantly increases. For this area (empirically selected) the median radiation difference is shown as blue dots instead of a blue line. A linear fit is calculated for the first part of the curves (red line). The fitted equations are shown in the top-left corner of the panels.





**Appendix A: COSMO-ART model configuration**

**Appendix B: Evaluation of RH vertical profiles**

**Appendix C: Hovmöller diagram of RH on 6–7 July 2016**

**Appendix D: Spatiotemporally-averaged profiles of aerosol mass concentration on 2–3 July 2016**

5 **Appendix E: Aerosol dry and wet diameters on 3 July 2016, 6 UTC**

**Appendix F: Realization-related cloud property differences on 2–3 July 2016**

**Appendix G: Hovmöller diagram of SSR, SLR and 2-m temperature differences on 6–7 July 2016**

**Appendix H: ALWC-radiation relationship on 6–7 July 2016**

*Acknowledgements.* The research leading to these results has received funding from the European Union 7th Framework Programme
10 (FP7/2007-2013) under Grant Agreement no. 603502 (EU project DACCIWA: Dynamics-aerosol-chemistry-cloud interactions in West
Africa). Thanks to the German Weather Service (DWD) for providing access to the ICON forecast data and to the Steinbuch Centre for
Computing (SCC) for providing the computational resources for the model realizations. The data analysis was done by using the software R
(2013).



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





**Table A1.** COSMO-ART model configuration used for this study.

| Characteristics | Description |
|---|---|
| Model version | COSMO5.1-ART3.1 |
| Time period | 2–3 July 2016, 6–7 July 2016 |
| Simulation domain | 9.0° W-4.4° E, 3.0° N-10.8° N |
| Grid mesh size | 2.5 km (0.0223°) |
| Vertical levels | 80 up to 30 km (28 in the lowest 1.5 km ASL) |
| Meteorological boundary and initial data | COSMO-ART (5 km grid mesh size using ICON operational forecasts from DWD) |
| Pollutant boundary and initial data | COSMO-ART (5 km grid mesh size using MOZART, 2017) <br> GlobCover (2009) landuse data <br> CCSM (2015) plant functional types |
| Cloud microphysics | Two-moment microphysics scheme (Seifert and Beheng, 2006) |
| Pollutant emissions | Mineral dust (online): Rieger et al. (2017) using HWSD (2012) <br> Sea salt (online): Lundgren et al. (2013) <br> DMS (online): using Lana et al. (2011) <br> BVOCs (online): Weimer et al. (2017) <br> Biomass burning (prescribed/online): Walter et al. (2016) using GFAS (CAMS, 2017) <br> Anthropogenic (prescribed): EDGAR (2010) <br> Gas flaring (prescribed): Deetz and Vogel (2017) |
| Aerosol dynamics | MADEsoot (Riemer et al., 2003; Vogel et al., 2009) <br> Secondary inorganic aerosol: ISORROPIA II (Fountoukis and Nenes, 2007) <br> Secondary organic aerosol: VBS (Athanasopoulou et al., 2013) |
| Chemical mechanisms | Gas phase chemistry: RADMKA (Vogel et al., 2009) |
| ALWC | Anthropogenic aerosol: ISORROPIA II <br> (Fountoukis and Nenes, 2007; Stokes and Robinson, 1966) <br> Sea salt: Lundgren et al. (2013) <br> Fresh soot: Riemer (2002) |
| Aerosol direct effect (ADE) | Vogel et al. (2009) |
| Aerosol indirect effect (AIE) | Warm phase: Bangert (2012) and Fountoukis and Nenes (2005) <br> Cold phase: Philipps et al. (2008) |



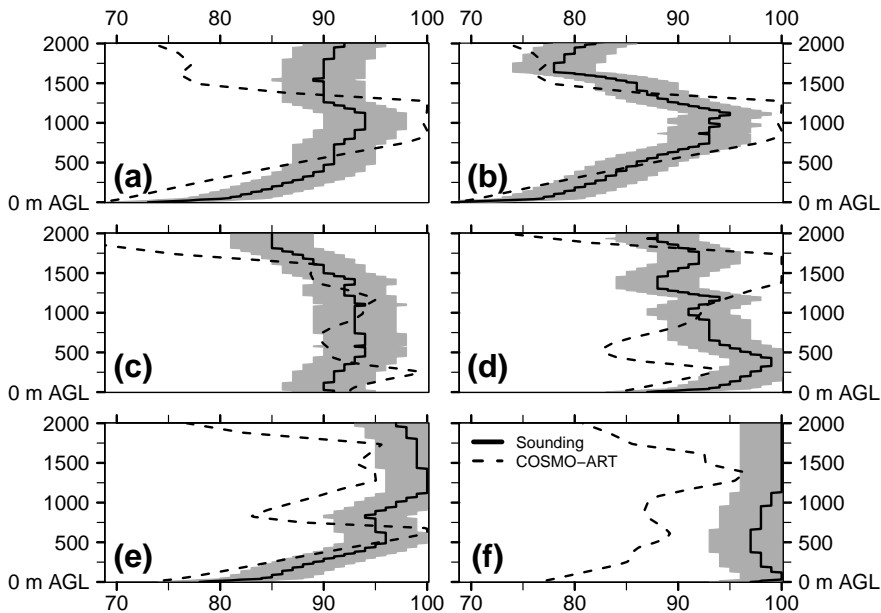

**Figure B1.** RH vertical profiles at Lamto (Ivory Coast) with respect to radiosoundings (black solid) and COSMO-ART (black dashed) on (a) 6 July 12 UTC, (b) 6 July 18 UTC, (c) 7 July 6 UTC, (d) 7 July 9 UTC, (e) 7 July 12 UTC and (f) 7 July 18 UTC. For the GRAW radiosondes an uncertainty of ±4 % are assumed (grey shading). The Lamto soundings are related to problems with reaching the 100 % RH (Andreas Fink, personal communication, 2018).

**Table F1.** Statistics of the cloud property differences (*Reference* minus *No-ALWC*, ICA) over Ivory Coast (7.5° W–3° W, 4–10° N) with respect to the time period 2 July 15 UTC to 3 July 15 UTC, including the median difference, the 25[th] and 75[th] percentile of the differences and the ratio of 75[th] percentile to the *Reference* average. The CDNC and the effective radius refer to the median in the lowest 1500 m AGL.

| Measure | Median | 25[th] percentile | 75[th] percentile | 75[th] percentile / *Reference* average |
|---|---|---|---|---|
| Total cloud water (g m$^{-2}$) | $-2.7 \cdot 10^{-4}$ | -27.9 | 22.6 | 0.37 |
| CDNC (cm$^{-3}$) | $-2.9 \cdot 10^{-9}$ | -59.1 | 54.7 | 0.79 |
| Effective radius ($\mu$m) | $1.8 \cdot 10^{-4}$ | -2.1 | 2.2 | 0.23 |





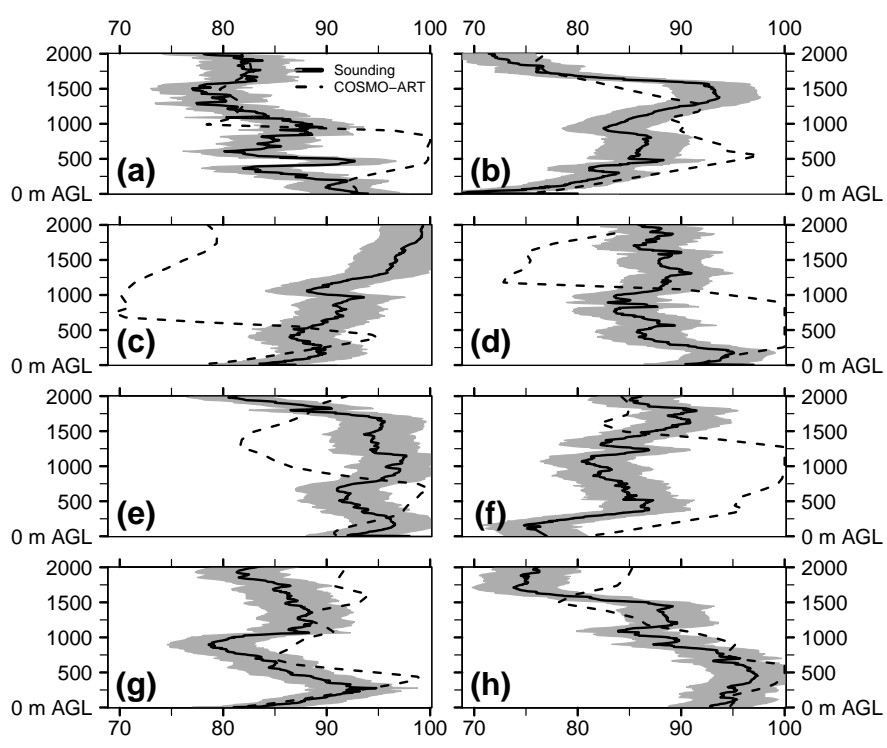

**Figure B2.** RH vertical profiles at Abidjan (Ivory Coast) with respect to radiosoundings (black solid) and COSMO-ART (black dashed) on (a) 2 July 4 UTC, (b) 2 July 10 UTC, (c) 2 July 16 UTC, (d) 2 July 23 UTC, (e) 3 July 4 UTC and (f) 3 July 10 UTC, (g) 3 July 16 UTC and (h) 3 July 23 UTC. For the Meteomodem radiosondes an uncertainty of $\pm 4$ % are assumed (grey shading).



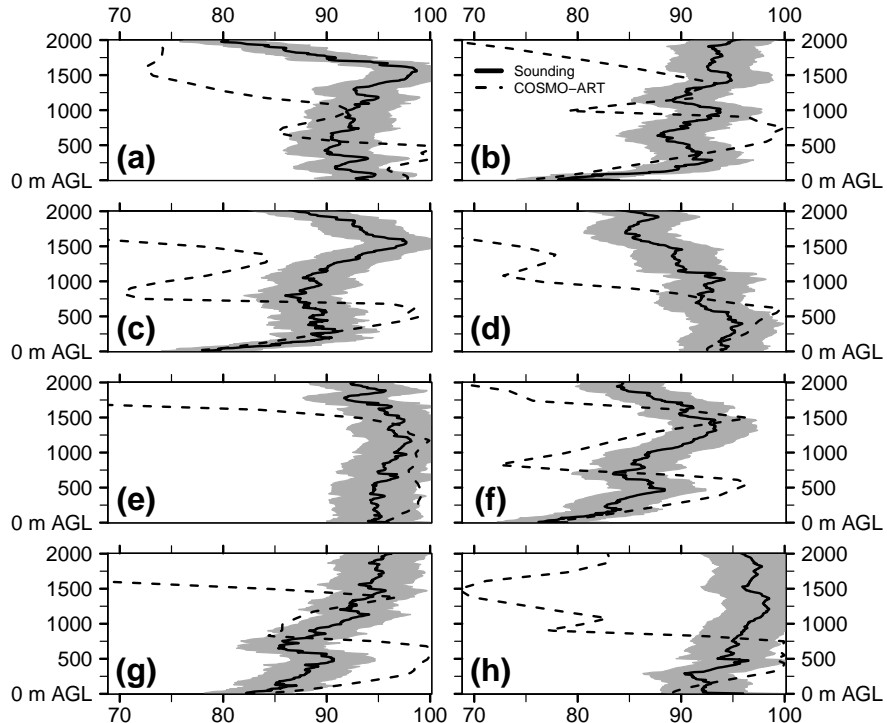

**Figure B3.** Same as for Figure B2 but for (a) 6 July 4 UTC, (b) 6 July 10 UTC, (c) 6 July 16 UTC, (d) 6 July 23 UTC, (e) 7 July 4 UTC, (f) 7 July 10 UTC, (g) 7 July 16 UTC and (h) 7 July 23 UTC.

**Table H1.** Radiation-ALWC relationship (W g$^{-1}$) based on the linear fits of Figure 15 including the time period 6 July 15 UTC to 7 July 15 UTC. The subdomain denotes whether the captured area is simultaneously cloudy (ICA) or cloud-free (OCA) in both realizations. *Original data* denote the slopes derived from Fig. 15 whereas *bootstrapping* refers to the median slopes of the 10.000 re-samples. The confidence intervals are derived by using the Gaussian approximation and $\alpha$=0.05 and the evaluation range provides the ALWC interval, which is used for the linear fitting.

| Radiation | Subdomain | Radiation-ALWC relationship (W g$^{-1}$) | | Evaluation range (g m$^{-2}$) |
|---|---|---|---|---|
| | | Original data | Bootstrapping | |
| SSR | ICA | -319 | -350±32 | 0.01–0.50 |
| SSR | OCA | -320 | -351±35 | 0.01–0.30 |
| Daytime SLR | ICA | -20 | -28±9 | 0.05–0.50 |
| Daytime SLR | OCA | -32 | -44±14 | 0.05–0.35 |
| Nighttime SLR | ICA | -14 | -20±5 | 0.05–0.45 |
| Nighttime SLR | OCA | -23 | -25±3 | 0.05–0.35 |


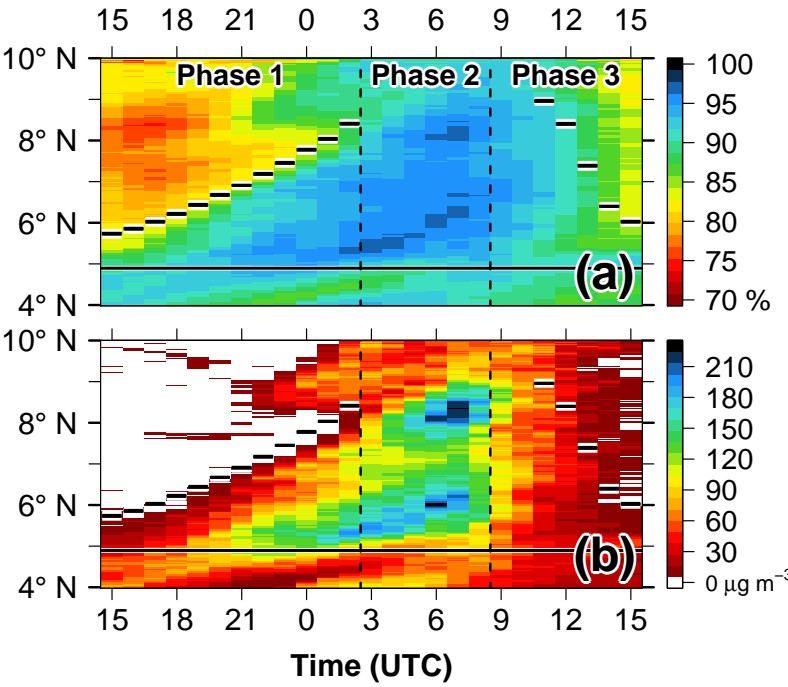

**Figure C1.** Hovmöller diagram of the median (a) RH (%) and (b) total ALWC ($\mu$g m$^{-3}$) in the lowest 1500 m AGL as zonal mean over Ivory Coast (7.5° W–3° W, 4–10° N) between 6 July 15 UTC and 7 July 15 UTC. The horizontal bars denote the zonal mean location of the 302 K isentrope at 250 m AGL, the horizontal solid line indicates the zonal mean coast line and the vertical dashed lines separate the three phases: *AI progression phase* (Phase 1), *Moist morning phase* (Phase 2) and *Daytime drying phase* (Phase 3).




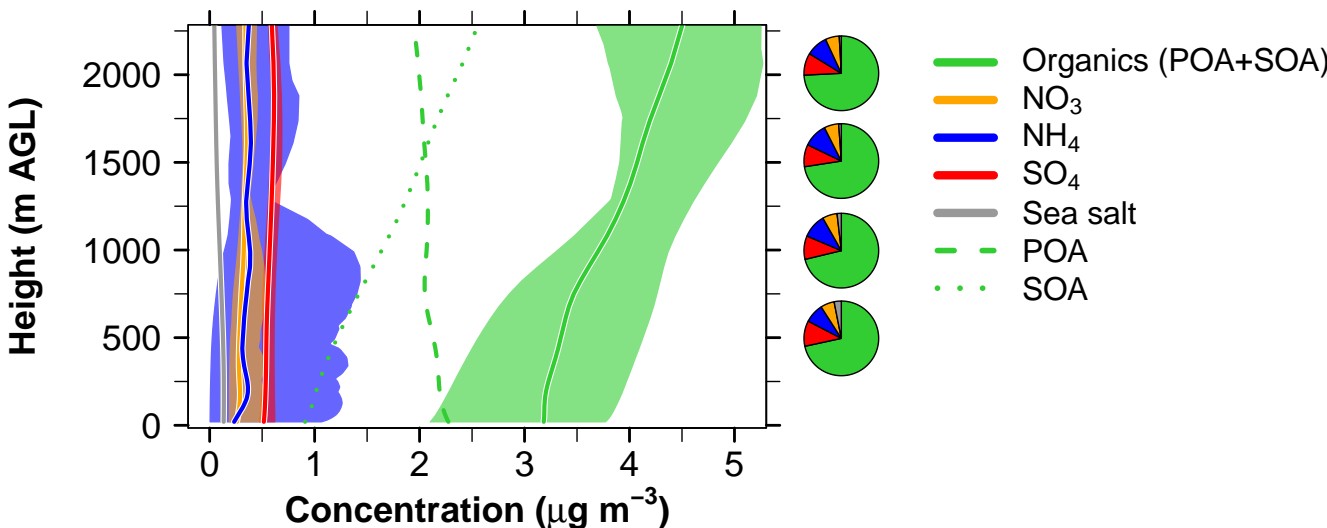

**Figure D1.** Vertical profiles (m AGL) of aerosol concentrations ($\mu$g m$^{-3}$) for the median over Ivory Coast (7.5° W–3° W, 4–10° N) with respect to the time period 2 July 15 UTC and 3 July 15 UTC. The colors refer to organics (POA+SOA; green solid line), NO$_3$ (orange solid line), NH$_4$ (blue solid line), SO$_4$ (red solid line) and sea salt (grey solid line). Additionally, POA and SOA are shown as dashed and dotted green lines, respectively. The shadings denote minima and maxima in the diurnal cycle mean profile and the pie charts on right hand side highlight the mean contribution of the single species to the total aerosol composition at 500, 1000, 1500 and 2000 m AGL.





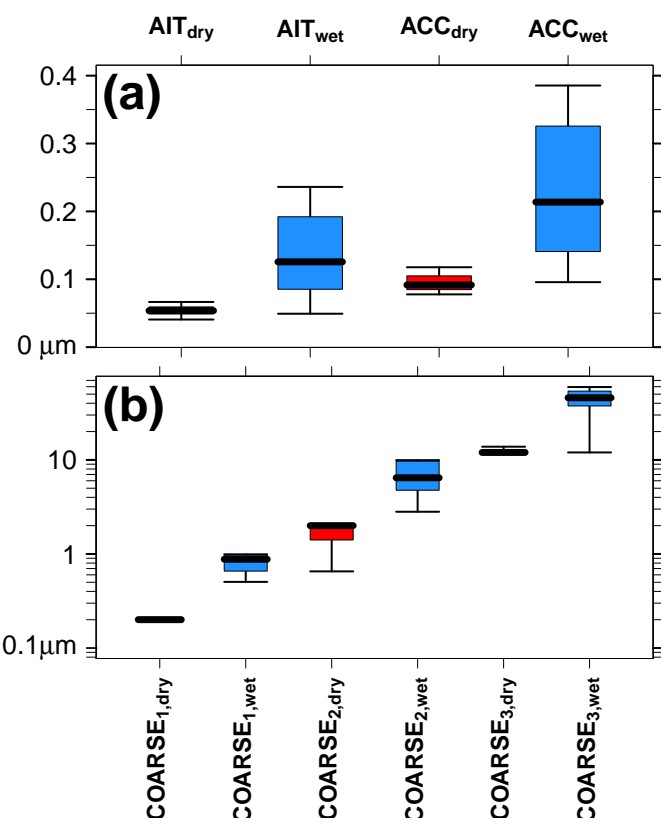

**Figure E1.** Boxplots of dry (red) and wet (blue) aerosol diameters ($\mu$m) for (a) AIT and ACC and (b) COARSE, splitted in the three COSMO-ART sea salt modes as median in the lowest 1500 m AGL over Ivory Coast (7.5° W–3° W, 4–10° N) on 3 July, 6 UTC. The whiskers span the data from the 2.5$^{\text{th}}$ to the 97.5$^{\text{th}}$ percentile (95 % of the data). Data outside of this range is not shown. Note the logarithmic scale in (b).



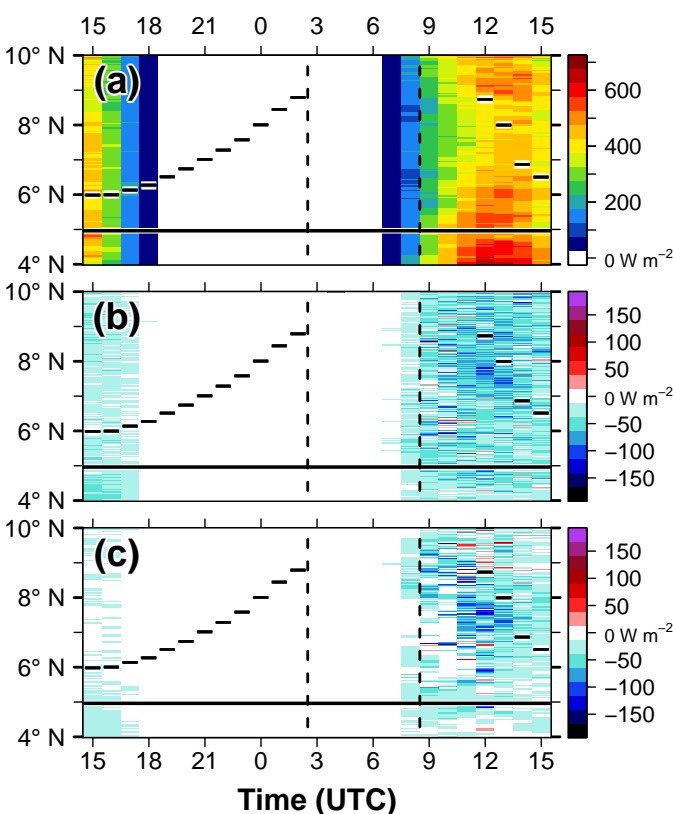

**Figure G1.** Hovmöller diagram of SSR (W m$^{-2}$) for (a) *Reference*, (b) *Reference* minus *No-ALWC* for ICA (c) *Reference* minus *No-ALWC* for OCA as zonal mean over Ivory Coast (7.5° W–3° W, 4–10° N) between 6 July 15 UTC and 7 July 15 UTC. The horizontal bars denote the zonal mean location of the 302 K isentrope at 250 m AGL of *Reference*, the horizontal solid line the zonal mean coast line and the vertical dashed lines the three phases introduced in Figure 3.





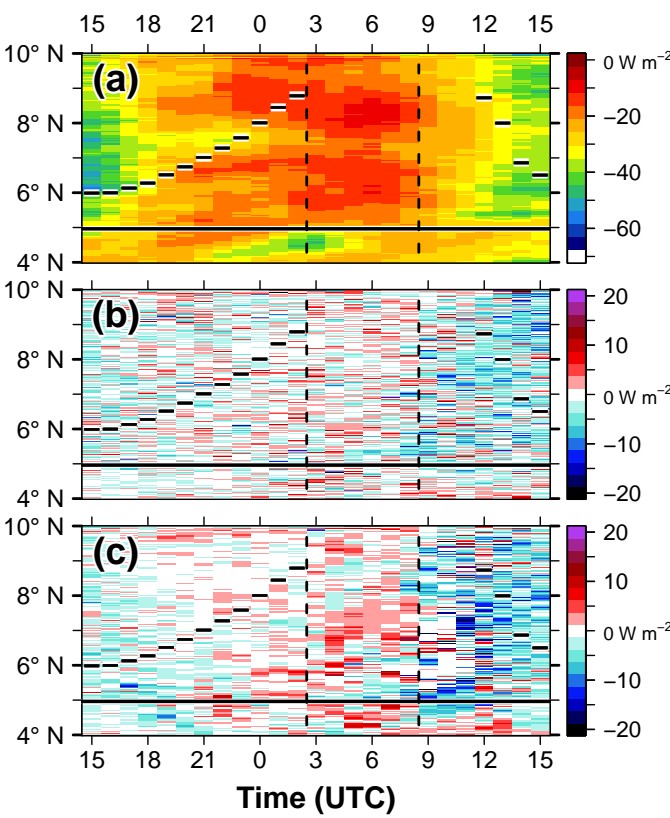

**Figure G2.** Same as for Fig. G1 but for the SLR. Positive (negative) values in (b) and (c) denote more outgoing longwave radiation in the *Reference* (*No-ALWC*) case.



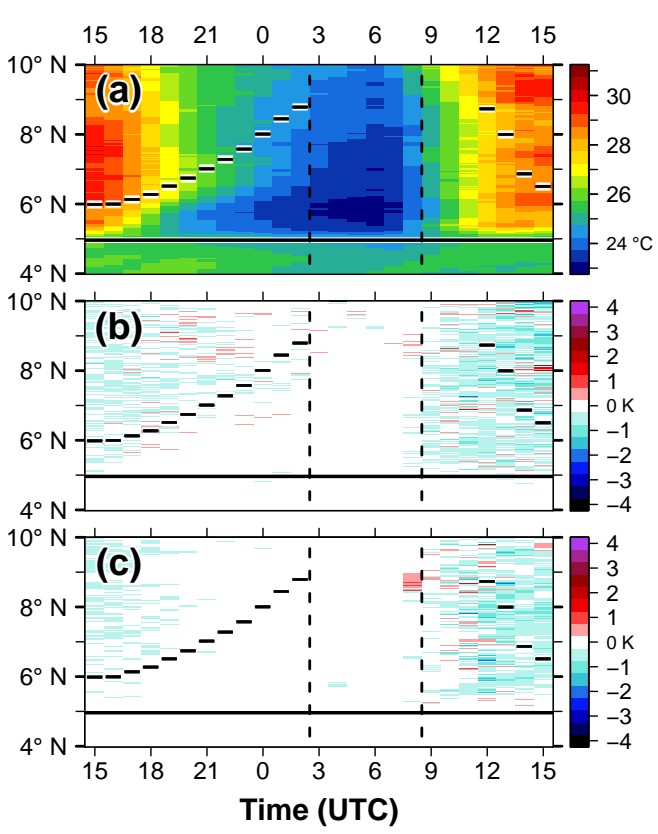

**Figure G3.** Same as for Fig. G1 but for 2-m temperature (°C) and 2-m temperature difference (K).




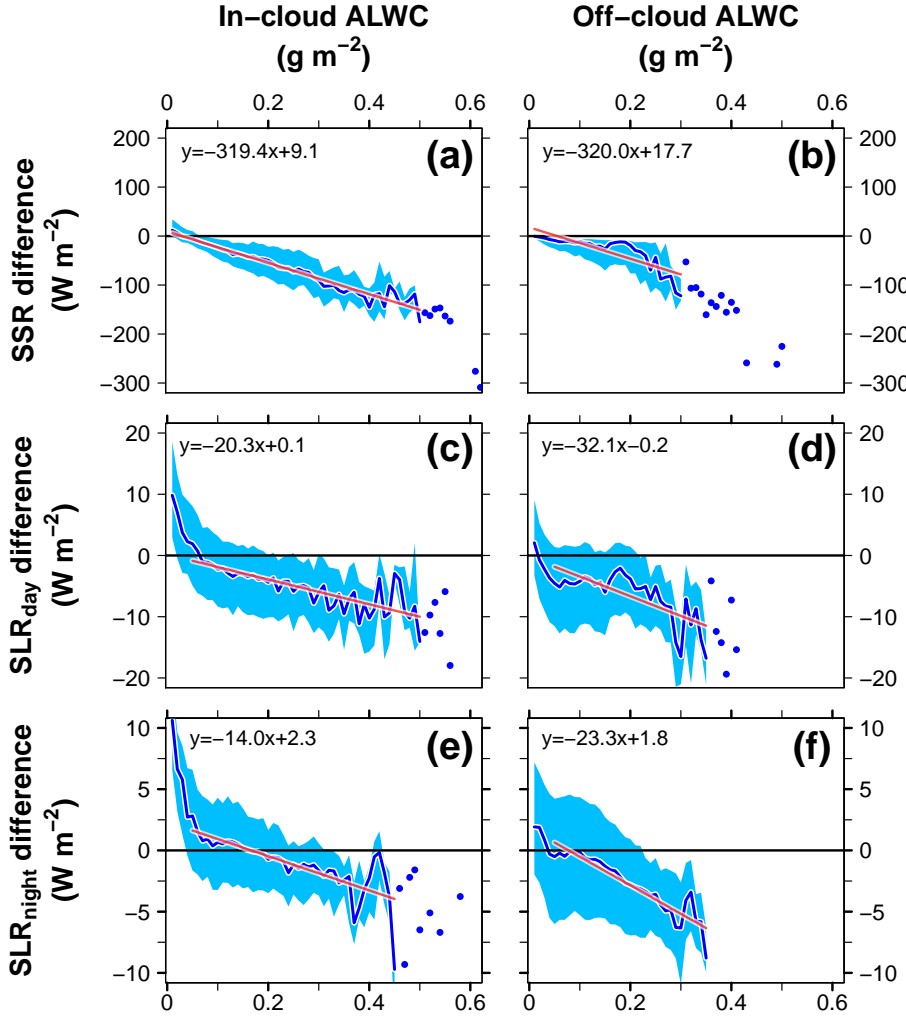

**Figure H1.** Relationship between the total column ALWC (g m$^{-2}$) and the radiation difference between *Reference* and *No-ALWC* (W m$^{-2}$) over Ivory Coast (7.5° W–3° W, 4–8° N) during the time period 2 July 15 UTC to 3 July 15 UTC. The data is clustered in areas that are simultaneously cloudy (left, ICA) or cloud free (right, OCA) in both realizations. The top panels show the SSR difference (6 July 15–18 UTC and 7 July 7–15 UTC), the middle panels the SLR daytime difference (same time period as SSR) and the bottom panels the SLR nighttime difference (2 July 19 UTC to 3 July 6 UTC). The ALWC values are clustered in bins with an increment of 0.01 g m$^{-2}$. For every bin the spatial median of the radiation difference is calculated (blue line). The envelope, spanned by the 25$^{th}$ and 75$^{th}$ percentile of the radiation difference, is shown as blue shading. For greater ALWC values the spread significantly increases. For this area (empirically selected) the median radiation difference is shown as blue dots instead of a blue line. A linear fit is calculated for the first part of the curves (red line). The fitted equations are shown in the top-left corner of the panels.