# Peer review of "Aerosol liquid water content in the moist southern West African monsoon layer and its radiative impact"

_Atmospheric Chemistry and Physics, 2018_

## Referee Comment (RC1) · Anonymous Referee #1 · 3 Jul 2018

Hygroscopic growth could alter the optical properties of aerosol. This manuscript reported the follow up simulation study based on Deetz et al. (2018) setup within the COSMO-ART modeling framework for a summer monsoon event in Southern West Africa and estimated the aerosol liquid water content (ALWC) and its impact on radiative transfer. The process was separated into three characteristic phases during commonly Atlantic Inflow event over this region to detailize the ALWC-radiation interactions. It was shown that the accumulation mode particles are the dominant contributor to aerosol liquid water and aerosol growth led to the increase of aerosol optical depth from 0.2 to 0.7. The increased aerosol optical depth can lead to around 20 W/m2 decrease in shortwave radiation. Bootstrapping technique was used to derive the linear

relationship between ALWC and radiation and found a stronger correlation for in-cloud conditions. This modeling study highlight the importance of including the relationship of RH dependency of aerosol optical depth in atmospheric model, which can significantly impact the local radiation balance, especially over moist tropical environment. The whole manuscript is well structured and the modeling discussion is adequate. I recommend publishing this work as a valuable component of the DACCIWA special issue in ACP after the authors address the following comments.

Page 1, Line 23: ALWC = aerosol liquid water content?

Page 4, Line 26-27: the "coarse modes of marine origin" should be (7-9) and the following "coarse modes of mineral origin" should be (10-12)?

Page 5, Line 5: ISORROPIA II does not include fresh soot for calculation. Did the model assume aged soot is internally mixed with sulfate in the calculation of optical properties and radiative transfer?

Page 5, Line 30-32. It is better to mark down the approximate area of "Ivory Coast" (7.5 W – 3W, 4N-10N, should be a subset of 2.5km modeling domain) in Figure 1(b) since nearly all the Figures follow on (e.g. Figure 2, . . .) are focus on this area.

Page 6, Line 21: Where is the geographic location of radiosounding site at "Lamto", please provide the locations in Figure 1. Also, look at the Figure B1-B3 in the Appendix, why there is no sounding comparison for location at "Lamto" for July 2-3. The radiosounding for RH vertical profiles at the two sites are not synchronized and with different time interval? Also, the Figure B2, may be due to the compress the the aspect ratio, the grey shading regions at certain place are not consistent with the description of uniformly 4% uncertainty assigned for radiosondes.

Page 7, line 22-23: ALWC was influenced by aerosol types and RH. Are the aerosol type and RH all the same in North China plain and southern West African, so they are comparable? The authors refer this study with China campaigns (e.g. HaChi,

[Figure]

PRIDE-PRD) heavily in the introduction section and the following discussion, maybe in some place in the introduction section, the author need to point out the similarity of this DACCIWA campaign with China campaigns such as aerosol loading, RH conditions, atmospheric oxidation capacity, cloud coverage.

Page 8, first paragraph: any explanation why OC dominate the aerosol mass composition? was it a biomass burning event? Also, for Figure D1, is the July 6-7 aerosol component vertical profiles similar to the July 2-3 shown here?

Page 8, Line 15. In contrast, AIT particles are lacking in size and COARSE particles are lacking in number.

Page 8, Line 30-31. Can you also provide the boxplots for median aerosol number concentrations for Aitken, accumulation and coarse mode in Figure E1?

Page 9, Line 8: the total water column is the full integration of model layer (e.g. 30km in Table S1) or below 1500m AGL that this study focused?

Page 9, Line 13-15: where is the location of the model realized NLLS and convective clouds in the fouced Ivory Coast region? In Figure 3 and Figure 4, the authors showed the double peak of ALWC during phase 2 period, one near coast and the another one in hilly terrain to the north. Are the peaks for ALWC at different locations also strictly correlated with the model simulated clouds?1

Page 10: Line 9-10. "... sharpen condition substantially decrease selected area", can you provide the percentage instead of the subjective description on simulated clouds grids versus non-cloud grids in the Ivory Coast area? From page 9, line 18-19, I may know only 3%-9% of total grids realized the clouds in July 2-3. So between the two sensitivity runs, the "ALWC" and "no-ALWC" case, how many percentage were excluded from further radiation analysis due to the model simulated the displacement of clouds?

Page 10, Line 28-29: where is the fixed SST value from COSMO-ART coming from?

Page 11, Line 24. The AOD is higher -> the difference of AOD is higher

[Figure]

Page 11, Line 33-34. In what percentage are the outliers for ALWC-radiation linear fitting (e.g. "less data, large spread, extra low ALWC ...")?

Page 12, Line 3-7. What the total size n for the linear fitting based on the grouping of ALWC versus radiation difference with the increment of 0.01 g m-2. If there existed similar linear reasonaliship either derived from observation or model from other regions, it is worthing mentioning here and discussing the possible reason for the difference considering during the DACCIWA campaign the aerosol components are dominated by OC (Figure D1) and the water uptake are most significant for coarse mode (Figure 6).

Page 14. Line 1-5. The authors mentioned before the RH underestimation may suggest the model derived ALWC value from this case study is the lower bound (Page 6, line 28-29), how it compared with the double counting of aerosol activate in the model, which tends to overestimate the AWLC, and the uncertainty for the corresponding radiation change calculation?

Page 18, Figure 4. The caption. "Same ass for Fig. 3"??

Page 34, in the row of "vertical levels", sometime in the main content the notation is "AGL" but here it is "ASL". make it consistent.

---

## Referee Comment (RC2) · Anonymous Referee #2 · 10 Jul 2018

Water uptake onto aerosol may increase the size of the aerosol population as well as their impact on global radiative budget. However, the models used nowadays do not take this effect in account properly. This study is based on simulations results to evaluate the impact of Aerosol Liquid Water Content (ALWC) on shortwave and longwave radiations over Southern West Africa. The authors try to estimate the effect of cloud presence, aerosol size and dynamical processes on ALWC. The manuscript is well written and definitely within the scopus of ACP. Therefore I recommend publishing this work after the authors address the following comments.

Major comments:

[Figure]

There are a lot of figures in this paper and I felt like most of them were not correctly described in the text. Indeed, each line drawn on a plot deserves at least a small explanation otherwise there is no need to plot it.

This kind of study is highly dependent on RH fields. In this manuscript, only profiles observed on July 2, 3, 5 and 6 2016 were compared to simulations results at two different locations (Lamto and Abidjan). Could you compare horizontal RH fields over West Africa for both periods?

Could you add more explanation about the dynamics of the Atlantic inflow? Indeed, sea breeze could be comparable to the Atlantic inflow but the occurrence time is not exactly the same. The AI front is moving inland during the night, which is quite unusual. During the night the ground temperature is getting colder in comparison to sea surface temperature. Therefore, I would rather imagine a land breeze. In few words, what is dynamically explaining this inflow?

P7 L 15-24 : The Aerosol Inflow involves an increase of RH a decrease of temperature but also brings different types of aerosols inland. You discuss the meteorological conditions that have for sure an influence on the ALWC but you never suggest that aerosol components may also have an impact. Section 3.3 (Impact of aerosol modes) : First, you should details the different types of aerosols that are predominant during each phase and the mean size distribution associated with each phase. Do you separate the aerosol modes in term of chemistry within your model? It seems, according to P8 L25, that coarse mode is only made of sea salt particles. How do you take into account dust then? The comparison with chinese field campaigns need to be clarified. Are the different types of aerosols similar in China and Africa? Did Chen et al. (2012) performed their measurements during the monsoon period?

P10 Section 4.2 : In this section, you are using 3 different figures to describe the effect of ALWC on the shortwave, longwave radiations and 2-m temperature. However, I felt like I did not have any explanations on what you observed. As an example, L16-18 'a

decrease in SSR can be observed when considering ALWC for ICA and OCA'. Could you explain why you have the same order of magnitude for OCA and ICA (where the RH should be higher)? 'A change in the cloud cover' âĞŠ the cloud is disappearing or strengthening? These are examples, but the entire section is written the same way.

According to your conclusions it seems that the cloud presence doesn't affect much the effect of ALWC on radiation. Could you provide anywhere in your manuscript the meteorological and aerosol size distribution differences between OCA and ICA?

Minor comments : Page 2 L10 : replace natrium by sodium

Page 3 L14-16 : I'm not sure I understand this sentence. You claim : " The RH increasingly affects the relationship between the amount of aerosol and the cloud droplet number concentration". I believe that larger RH could involve more or larger cloud droplets. These results are not from 2015...

Page 3 L28 : I believe that there were no GF measured during AMMA.

P5 L4 : Could 5% of the mass concentration of soot particle be defined considerable ?

P5 L16-17 : 'Furthermore, . . . process studies'. I do not understand this sentence. The undisturbed moosoon condition favor NLLS presence? Also, NLLS is not defined in the acronym list.

P5 L28 : please remove 'by a a decrease'

P9 section 3.4 : Is this AOD within cloud ? Are you talking about interstitial aerosols? Then the clouds are just considered as a vector for RH increase? There are numerous studies that have shown the contribution of the ALWC to the total AOD (Brock et al., 2015 and 2016; Crumeyrolle et al., 2014; Beyersdorf et al., 2016; Orozco et al. 2016; Eck et al. 2014).

P12 L22 : please remove 'The' : 'on THE one hand'

P13 L3 : please replace 'AI affected' - 'AI affects'

Figure 4 : I'm sure this is a typo : ' same ass ' Relative ALWC should be a proxy for the hygroscopicity of aerosols right ? If yes then it needs to be stated somewhere. And you should present mean aerosol size distribution before this figure for the different phases.

Figure 5 : You should add on the different figures 'TOTAL' , 'AIT', 'ACC' and 'COARSE'

Figure 6 : Could you add the RH on this figure ?

Figure 9/10/11 : Could you add on the figure ICA and OCA . I'm sure that will also be clearer if there is REF and REF-No_ALWC

―――――――――――――――――

---

## Referee Comment (RC3) · Anonymous Referee #2 · 10 Jul 2018

It's not clear if the radiative effect of the ALWC is calculated only from an increase of the aerosol size or if the refractive index is changing with the water uptake ?

---

## Author Comment (AC1) · 4 Aug 2018

Answer to Referee #1 Konrad Deetz 25 July 2018

Dear Referee (Atmospheric Chemistry and Physics),

thank you for your report from 3 July 2018. We have accounted for the comments and suggestions in the revised manuscript version. Please find our replies (marked with #) to the individual comments in the following.

Sincerely, Konrad Deetz on behalf of all coauthors

Referee comments: (0) Hygroscopic growth could alter the optical properties of

aerosol. This manuscript reported the follow up simulation study based on Deetz et al. (2018) setup within the COSMO-ART modeling framework for a summer monsoon event in Southern West Africa and estimated the aerosol liquid water content (ALWC) and its impact on radiative transfer. The process was separated into three characteristic phases during commonly Atlantic Inflow event over this region to detailize the ALWC-radiation interactions. It was shown that the accumulation mode particles are the dominant contributor to aerosol liquid water and aerosol growth led to the increase of aerosol optical depth from 0.2 to 0.7. The increased aerosol optical depth can lead to around 20 W/m2 decrease in shortwave radiation. Bootstrapping technique was used to derive the linear relationship between ALWC and radiation and found a stronger correlation for in-cloud conditions. This modeling study highlight the importance of including the relationship of RH dependency of aerosol optical depth in atmospheric model, which can significantly impact the local radiation balance, especially over moist tropical environment. The whole manuscript is well structured and the modeling discussion is adequate. I recommend publishing this work as a valuable component of the DACCIWA special issue in ACP after the authors address the following comments.

(1) Page 1, Line 23: ALWC = aerosol liquid water content?

**We have changed the manuscript accordingly.**

(2) Page 4, Line 26-27: the "coarse modes of marine origin" should be (7-9) and the following "coarse modes of mineral origin" should be (10-12)?

**We have changed the manuscript accordingly.**

(3) Page 5, Line 5: ISORROPIA II does not include fresh soot for calculation. Did the model assume aged soot is internally mixed with sulfate in the calculation of optical properties and radiative transfer?

**Yes, fresh soot is not included in ISORROPIA II. In COSMO-ART it is therefore handeled separately (as denoted on p. 4 l. 3). And yes, as soon as the soot is treated as**

[Figure]

aged, it is an internal mixture within ISORROPIA II and with respect to the calculation of optical properties and radiative transfer.

(4) Page 5, Line 30-32. It is better to mark down the approximate area of "Ivory Coast" (7.5 W – 3W, 4N-10N, should be a subset of 2.5km modeling domain) in Figure 1(b) since nearly all the Figures follow on (e.g. Figure 2, : : :) are focus on this area.

**We agree on that and have changed the manuscript (figure and figure caption) accordingly.**

(5) Page 6, Line 21: Where is the geographic location of radiosounding site at "Lamto", please provide the locations in Figure 1. Also, look at the Figure B1-B3 in the Appendix, why there is no sounding comparison for location at "Lamto" for July 2-3. The radiosounding for RH vertical profiles at the two sites are not synchronized and with different time interval? Also, the Figure B2, may be due to the compress the the aspect ratio, the grey shading regions at certain place are not consistent with the description of uniformly 4% uncertainty assigned for radiosondes.

**We added Lamto as a magenta dot in Figure 1b. For Lamto, no sounding data is available for 2-3 July. Indeed, the soundings of Lamto and Abidjan are not launched at the same times and with different time intervals. We double-checked the shaded area enveloping the uncertainty of +-4 % relative humidity. This is correct. The shading just appears inhomogeneous when the black line is rather horizontal.**

(6) Page 7, line 22-23: ALWC was influenced by aerosol types and RH. Are the aerosol type and RH all the same in North China plain and southern West African, so they are comparable? The authors refer this study with China campaigns (e.g. HaChi, PRIDE-PRD) heavily in the introduction section and the following discussion, maybe in some place in the introduction section, the author need to point out the similarity of this DACCIWA campaign with China campaigns such as aerosol loading, RH conditions, atmospheric oxidation capacity, cloud coverage.

**This is an interesting question. We try to elaborate this by relating to the study of Bian et al. (2014) as you have proposed. When focusing on the study of Bian et al. (2014), the observations are related to the time period July-August 2009 and focusing on the chinese provinces Shandong, Hebei, Pekin and Tianjin. The climate in this area is in between humid subtropical and humid continental Summers are hot and rainy with temperatures around 24-28 °C in July with the precipitation maximum in summer via influences from the monsoon. A qualitative analysis of Terra Modis satellite images (of course only one overfly per day) revealed that in the 62 d period of July-August 2009 Shandon was fully covered by clouds on 55 d and partly covered by clouds on 7 days. Therefore the weather conditions during the DACCIWA campaign and HaChi campaign are very similar. Both studies focus on the NH summer. Both areas are located in the NH summer monsoon area with high temperatures and are very frequent covered by clouds. The measurement site for the study of Bian et al. (2014) is Wuqing. For this location, Liu et al. (2011) [Figure 3] shows measurements of temperature and relative humidity for July-August 2009. Temperature variations are between 20 °C and 32 °C. Relative humidity variations are between 40 % (mostly 60%) and 95 %. The latter is similar to what is modeled for southern West Africa (Fig. 3 in our manuscript) and to what was observed in southern West Africa at Save supersite (Kalthoff et al., 2018, Fig. 3).**

Wuqing is about 90 km away from the Gulf of Bohai. So also HaChi focuses on the area near the coast. Wuqing is surrounded by large cities (Peking (80 km away, 21.5 million inhabitants, megacity), Langfang (30 km away, 4.4 million inhabitants), Tianjin (40 km away, 15.5 million inhabitants, megacity), Tangshan (100 km away, 7.6 million inhabitants)). Also southern West Africa has several large cities especially near the coast. However, the populations are generally smaller but on the same order of magnitude (Lagos: 13.7 million inhabitants, Abidjan: 5 million inhabitants). Based on MODIS observations, Bian et al. (2014) show that the averaged AOD values are generally above 0.6 in the research area and 0.7 above Wuqing. For the DACCIWA region we found averaged MODIS AOD values of 0.4-0.7, slightly smaller to what was observed

in the HaChi region. However, the validity over land is limited because southern West Africa is virtually always covered by clouds, restricting the observations to a few days.

Based on these findings we came to the conclusion that the general meteorological and aerosol conditions are similar for HaChi and DACCIWA and therefore allow a qualitative comparison e.g. of the ALWC values between both sites.

We added the following passage in the conclusions to account for your remark: "HaChi and DACCIWA both focus on the northern hemispheric monsoon season, capture coastal areas that are frequently covered by clouds, have similar temperature and relative humidity conditions (Liu et al., 2011; Kalthoff et al., 2018) as well as similar aerosol loadings (Bian et al. (2014); Deetz et al. (2018a), allowing for a qualitative comparison of modeled ALWC with measurements during HaChi."

(7) Page 8, first paragraph: any explanation why OC dominate the aerosol mass composition? was it a biomass burning event? Also, for Figure D1, is the July 6-7 aerosol component vertical profiles similar to the July 2-3 shown here?

**The aerosol mass composition is subject to current research in the DACCIWA research community. Therefore the main outcomes with respect to this question are not yet available/published. However, also the DACCIWA observations (e.g. aircraft measurements) show this dominance of organic carbon (e.g. Flamant et al., 2018). Biomass burning is an important source of OC and likely is responsible for the dominance of OC over Ivory Coast. Based on the experience we obtained with COSMO-ART during our two month (June-July 2016) of forecasting the atmospheric composition (with coarser grid mesh size), we observed that the biomass burning plumes over the Gulf of Guinea (coming from central Africa) frequently swash into the DACCIWA domain. To account for your remark, we repeated the composition analysis for 6-7 July 2016. The corresponding plot (Review-figure-1) is given as appendix to our review answer. For the non-OA, the situation is comparable with 2-3 July but OA is about twice as high compared to 2-3 July with a more distinct vertical gradient, indicating a stronger**

influence of biomass burning.

(8) Page 8, Line 15. In contrast, AIT particles are lacking in size and COARSE particles are lacking in number.

**We have changed the manuscript accordingly.**

(9) Page 8, Line 30-31. Can you also provide the boxplots for median aerosol number concentrations for Aitken, accumulation and coarse mode in Figure E1?

**We have changed Figure E1 and the manuscript accordingly. Now the panels (a) and (c) show the median aerosol number concentrations for Aitken, accumulation and coarse mode in addition to the aerosol diameters (b,d). The revised figure is added as appendix (Review-figure-4).**

(10) Page 9, Line 8: the total water column is the full integration of model layer (e.g. 30km in Table S1) or below 1500m AGL that this study focused?

**Yes, in this case the full integration of model layer is considered and not just the lowest 1500 m. This is done on purpose because the total cloud water column is a widely used measure for the quantification of clouds and with this figure we want to provide some guide values to allow for comparison between the water contribution from clouds and the water contribution from aerosol. Furthermore, Figure 7 is the basis for Figure 8 and in Figure 8 we also analyze the contribution of the in-cloud AOD to the total AOD. Since the total AOD is related to the total vertical column, it is necessary to focus on the total vertical column in the model to ensure consistency.**

(11) Page 9, Line 13-15: where is the location of the model realized NLLS and convective clouds in the fouced Ivory Coast region? In Figure 3 and Figure 4, the authors showed the double peak of ALWC during phase 2 period, one near coast and the another one in hilly terrain to the north. Are the peaks for ALWC at different locations also strictly correlated with the model simulated clouds?1

**We attached Review-figure-2 to emphasize the location of clouds over Ivory Coast**

and the total DACCIWA domain in general. The figure shows an overview of the low-level cloud temporal evolution between 2 July 21 UTC and 3 July 10 UTC (a-f). Blue shading denotes low-level clouds via the existence of cloud water in the lowest 1.3 km AGL. Brown shading indicates the topography above 250 m ASL. The arrows show the wind speed (m s-1, scale is given below) and direction at 250 m AGL. For 21 UTC and 23 UTC the Atlantic Inflow front is shown in red. From a-c a clear separation between the cloud band directly behind the Atlantic Inflow front and at the coast is visible. This figure is published in Deetz (2018a). The ALWC is primarilly correlated with the relative humidity, therefore cloudy areas (with a presence of sufficient amounts of aerosol, which is fulfilled over the entire DACCIWA domain) are areas with the highest amounts of ALWC. Review-figure-3 shows the ALWC at 500 m AGL (where we can find the NLLS) over land (for the entire DACCIWA domain) on 3 July 6 UTC. (a) Total ALWC (mg m-3, shading) and RH of 95% (black contour) and (b) pie chart of the ALWC contribution from the single aerosol modes (%) to the total ALWC in (a). For the entire DACCIWA domain highest ALWC values can be found in areas with highest relative humidities (location of the NLLS). This figure is also published in Deetz (2018a).

(12) Page 10: Line 9-10. "... sharpen condition substantially decrease selected area", can you provide the percentage instead of the subjective description on simulated clouds grids versus non-cloud grids in the Ivory Coast area? From page 9, line 18-19, I may know only 3%-9% of total grids realized the clouds in July 2-3. So between the two sensitivity runs, the "ALWC" and "no-ALWC" case, how many percentage were excluded from further radiation analysis due to the model simulated the displacement of clouds?

**We calculated the ratio a/b with (a) the number of gridboxes which are related to clouds in both realizations by restricting to gridboxes with a total cloud water difference below 0.1 g m-2 (masking cloud displacement) (b) and the number of gridboxes which are related to clouds in both realizations without any restrictions (by ignoring cloud displacement). This ratio is between 0.04 and 0.18 in the 25 hour period with a median**

of 0.076. So on average only 7.6 % of the cloud grid points (clouds in both realizations) can be used for the radiation analysis. We adapted the corresponding sentence in the manuscript as follows: "Consider that the sharpened condition substantially decreases the selected area (on average only 7.6 % of the cloudy area can be considered) and therefore makes the results less representative for the cloudy area."

(13) Page 10, Line 28-29: where is the fixed SST value from COSMO-ART coming from?

**The fixed SST is coming from the driving model ICON. For ICON, the SST fields are derived daily at 0 UTC based on observations. A detailed description of the handling of the SST in COSMO can be found in the "COSMO Documentation Part III - Data assimilation" (http://www.cosmo-model.org/content/model/documentation/core/cosmoAssim.pdf) at page 89f.**

(14) Page 11, Line 24. The AOD is higher -> the difference of AOD is higher

**We have changed the manuscript accordingly.**

(15) Page 11, Line 33-34. In what percentage are the outliers for ALWC-radiation linear fitting (e.g. "less data, large spread, extra low ALWC ...")?

**The following tables summarize the percentages of ALWC data that are not included in the linear fitting (red curves in Fig. 15 (2-3 July) and Fig. H1 (6-7 July)).**

2-3 July 2016 (Fig. 15): see Table presented in Review-figure-5

6-7 July 2016 (Fig. H1): see Table presented in Review-figure-6

This analysis shows very similar results when comparing 2-3 July and 6-7 July underlining some robustness in these characteristics. The upper outliers are generally noncritical and negligible (never greater than 0.5%). Lower outliers are only relevant when focusing on longwave radiation and in-cloud areas because there a nonlinear behavior is obvious for which we have no explanation. In this case about one-fifth of

the data is not considered. We added/adapted the following passage in the manuscript to meet your concerns: "The fitting omits bins with large ALWC (less data and large spread). A detailed analysis revealed that not more than 0.5 % of the data are omitted. Figure 15c and Figure 15e show a nonlinear behavior for low ALWC. Therefore also these parts are omitted in the linear fitting. This affects 3.5-23.3 % of the data."

(16) Page 12, Line 3-7. What the total size n for the linear fitting based on the grouping of ALWC versus radiation difference with the increment of 0.01 g m-2. If there existed similar linear reasonaliship either derived from observation or model from other regions, it is worthing mentioning here and discussing the possible reason for the difference considering during the DACCIWA campaign the aerosol components are dominated by OC (Figure D1) and the water uptake are most significant for coarse mode (Figure 6).

**To the first part of your question: In the style of our tables of remark (15) we show again the two tables that now include the total number n of gridboxes that are used for the linear fitting. n_max is the maximum number for day and night spanned by the dimensions lon x lat x hours.**

2-3 July 2016 (Fig. 15): see Table presented in Review-figure-7

6-7 July 2016 (Fig. H1): see Table presented in Review-figure-8

To the second part of your question: As far as we know, our study is the first approach assessing the linear relationship between ALWC and the radiation difference. We don't have opportunities for a comparison with observations or model results from other regions. (A prerequisite for a comparison of our results with model results from other regions is the availability of a model run that excludes the ALWC effect in the radiative transfer calculations that can be compared with a reference run. From our knowledge, this is not available from other research groups.) Zieger et al. (2017) made an approach with a global model to underline how the hygroscopicity of sea salt affects the AOD (and with that the radiative transfer which is not shown in that work). However, we

have serious doubts that a global model is able to appropriately consider the aerosol growth due to water uptake and their impacts on radiation. Nevertheless we added a reference of this work in our introductory section as follows: "Ziegler et al. (2017) assess the effect of hygroscopicity of sea salt on AOD with a global model approach. They modeled latitudinal averaged reductions in the AOD of up to 14 % when reducing the hygroscopicity of sea salt from 1.5 to 1.1." It is not unusual that OC is the dominating aerosol component, especially when regions are affected by locally emitted or long-range transported biomass burning plumes. E.g. Brito et al. (2014) characterize the ground-based aerosol during the South American Biomass Burning Analysis (SAMBBA) field experiment and found that OC is the dominating aerosol in the submicron size range. With respect to the significant water uptake of the coarse mode it has to be considered that in our radiation analysis the coarse mode only consists of sea salt. Generally, the coarse mode in COSMO-ART consists of sea salt, mineral dust and coarse mode anthropogenic particles. But the latter two are not related to ALWC in COSMO-ART. It is not a new finding that sea salt is extremely hygroscopic. Sea salt aerosol particles take up significant amounts of water at RH < 75%, due to the presence of the highly hygroscopic salts of $Ca^{2+}$ and $Mg^{2+}$ (Ziegler et al., 2017). Therefore we have expected most significant water uptake with respect to the coarse mode.

(17) Page 14. Line 1-5. The authors mentioned before the RH underestimation may suggest the model derived ALWC value from this case study is the lower bound (Page 6, line 28- 29), how it compared with the double counting of aerosol activate in the model, which tends to overestimate the AWLC, and the uncertainty for the corresponding radiation change calculation?

**The comparison of the modeled RH with soundings at Abidjan and Lamto (Figure B1-B3) indicate that COSMO-ART tends to underestimate the RH, although there is no systematic bias. This is a source of uncertainty for the calculation of ALWC and the radiative transfer. However, it has to be considered that the increase in water uptake**

is most sensitive to RH in the narrow range of RH >95 % and less sensitive for RH below 95 %. Therefore, potential deviations should not be overrated. The conception in COSMO-ART, not to remove the activated aerosol from the aerosol population, is done by reason. Model tests in the past that remove the aerosol after activation leads to a very fast (unrealistic) cleaning of the atmosphere. But conception of not removing the activated aerosol from the aerosol population does not lead to an overestimation of the ALWC. Instead it is the consideration of two different aspects: (a) Aerosol that take up water, (b) A cloud droplet or ice crystal that has an aerosol particle (CCN/IN) inside. The activated aerosol particle is a cloud droplet (or ice crystal) and the radiative interaction is only related to its quality being a cloud droplet (the negligible small aerosol particle and its ALWC is not considered when we talk about the interaction between cloud droplet and radiation). On the other hand we have the aerosol in the aerosol population that can take up water when it is hygroscopic. In this case there is an interaction between the aerosol particle (combination of aerosol and ALWC) and the radiation. Therefore we expect that we do not per se overestimate the ALWC with our model concept. But of course, we see uncertainties in the corresponding radiative transfer calculations. With our existing model system and the model realizations we have conducted for this study it is not possible to quantify these uncertainties or to set them in relation to the uncertainty that comes from deviations in the RH.

(18) Page 18, Figure 4. The caption. "Same ass for Fig. 3"??

**We have changed the manuscript accordingly.**

(19) Page 34, in the row of "vertical levels", sometime in the main content the notation is "AGL" but here it is "ASL". make it consistent.

**We have changed the manuscript accordingly.**

Additional References: Flamant et al. (2018): THE DYNAMICS–AEROSOL–CHEMISTRY–CLOUD INTERACTIONS IN WEST AFRICA FIELD CAMPAIGN Overview and Research Highlights, BAMS, pp. 83-104,

https://journals.ametsoc.org/doi/pdf/10.1175/BAMS-D-16-0256.1

Zieger et al. (2017): Revising the hygroscopicity of inorganic sea salt particles, Nature communications, Vol. 8, Article number: 15883, https://www.nature.com/articles/ncomms15883
[Figure]

[Figure]

**Fig. 1.** Vertical profiles (m AGL) of aerosol concentrations (ug m-3) for the median over Ivory Coast (7.5° W–3° W, 4–10° N) with respect to the time period 6 July 15 UTC and 7 July 15 UTC.

[Figure]

**Fig. 2.** Overview of the low-level cloud temporal evolution between 2 July 21 UTC and 3 July 10 UTC (a-f). Blue shading denotes low-level clouds via nonzero cloudwater below 1.3 km AGL.

[Figure]

[Figure]

**Fig. 3.** ALWC at 500 m AGL over land on 3 July 6 UTC. (a) Total ALWC (mg m-3, shading) and RH of 95% (black contour) and (b) pie chart of the ALWC.

**Fig. 4.** Boxplots of (a) aerosol number density (cm-3) and (b) dry (red) and wet (blue) aerosol diameters (um) for AIT and ACC and boxplots of (c) aerosol number density (cm-3) and (d) dry and wet diameter.

```
2-3 July 2016 (Fig. 15):
            In-loud ALWC (ICA)         |      Off-cloud ALWC (OCA)
* * *
         lower outlier    upper outlier | lower outlier    upper outlier
SW day        -                0.4%     |      -                0.2%
LW day      23.3%              0.4%     |      -                0.05%
LW night     3.5%              0.5%     |      -                0.02%
```

**Fig. 5.** Table summarizing the percentages of ALWC data that are not included in the linear fitting (red curves in Fig. 15 (2-3 July)).

```
6-7 July 2016 (Fig. H1):
             In-loud ALWC (ICA)        |      Off-cloud ALWC (OCA)
* * *
          lower outlier    upper outlier  |  lower outlier    upper outlier
SW day        -               0.1%        |      -               0.2%
LW day      16.7%             0.1%        |      -               0.05%
LW night     3.5%             0.2%        |      -               0.06%
```

**Fig. 6.** Table summarizing the percentages of ALWC data that are not included in the linear fitting (red curves in Fig. H1 (6-7 July)).

```
2-3 July 2016 (Fig. 15):
n_max_day  = 472680
n_max_night= 436320
                 In-loud ALWC (ICA)  |  Off-cloud ALWC (OCA)
* * *
                        n            |            n
SW day               16560          |          55120
LW day               12687          |          55230
LW night              9310          |         114402
```

**Fig. 7.** Table showing the total number n of gridboxes that are used for the linear fitting for 2-3 July 2016 (Fig. 15).

```
6-7 July 2016 (Fig. H1):
n_max_day  = 472680
n_max_night= 436320
            In-loud ALWC (ICA)  |  Off-cloud ALWC (OCA)
* * *
                  n             |           n
SW day            17086         |           48518
LW day            14236         |           48583
LW night          8903          |           109723
```

**Fig. 8.** Table showing the total number n of gridboxes that are used for the linear fitting for 6-7 July 2016 (Fig. H1).

---

## Author Comment (AC2) · 4 Aug 2018

Answer to Referee #2a Konrad Deetz 25 July 2018

Dear Referee (Atmospheric Chemistry and Physics),

thank you for your report from 10 July 2018. We have accounted for the comments and suggestions in the revised manuscript version. Please find our replies (marked with #) to the individual comments in the following.

Sincerely, Konrad Deetz on behalf of all coauthors

Referee comments: (0) Water uptake onto aerosol may increase the size of the aerosol

population as well as their impact on global radiative budget. However, the models used nowadays do not take this effect in account properly. This study is based on simulations results to evaluate the impact of Aerosol Liquid Water Content (ALWC) on shortwave and longwave radiations over Southern West Africa. The authors try to estimate the effect of cloud presence, aerosol size and dynamical processes on ALWC. The manuscript is well written and definitely within the scopus of ACP. Therefore I recommend publishing this work after the authors address the following comments.

(1) There are a lot of figures in this paper and I felt like most of them were not correctly described in the text. Indeed, each line drawn on a plot deserves at least a small explanation otherwise there is no need to plot it.

**The figures are necessary to transport our findings to the reader. The number of figures increased because we actively decided to repeat some of the pivotal figures also for the time period 6-7 July 2016 in the appendix (in addition to our main focus 2-3 July 2016). This is done to support our findings, making them more robust within our limited capacity to run further computationally expensive model realizations. We think that all figures are described in detail. If you have the feeling that a figure is not correctly described please indicate which passage has to be revised.**

(2) This kind of study is highly dependent on RH fields. In this manuscript, only profiles observed on July 2, 3, 5 and 6 2016 were compared to simulations results at two different locations (Lamto and Abidjan). Could you compare horizontal RH fields over West Africa for both periods?

**We agree, RH is the predominant factor for ALWC. We are convinced RH profiles from soundings are appropriate to evaluate the modeled RH. Radiosounding is one of the most accurate measurement techniques for quantifying RH. Horizontal fields of observed RH are not available from DACCIWA observations. Also remote sensing does not provide horizontal RH fields but statements about the total column water. However, remote sensing is extremely limited over SWA due to the frequent cloud cover.**

[Figure]

(3) Could you add more explanation about the dynamics of the Atlantic inflow? Indeed, sea breeze could be comparable to the Atlantic inflow but the occurrence time is not exactly the same. The AI front is moving inland during the night, which is quite unusual. During the night the ground temperature is getting colder in comparison to sea surface temperature. Therefore, I would rather imagine a land breeze. In few words, what is dynamically explaining this inflow?

**We suggest, the two counteracting effects "pressure difference" and "turbulence difference" determine the AI front and its propagation. During day the land is subject to stronger heating than the Gulf of Guinea, leading to stronger turbulence over land. The turbulence mixes the horizontal momentum of the monsoon flow vertically, impeding the monsoon flow and establishing a frontal structure near the coast. In the evening, the turbulence over land decreases allowing the pressure difference (pressure gradient in direction land-sea) to overcome the effects from turbulence. The front starts to penetrate inland, transporting the post-frontal air characteristics (cool air, low-level jet) inland. Therefore during night the monsoon flow (directed from ocean to land) overcompensates the land breeze that we would expect in the classical land-sea breeze concept. Please also refer to our companion paper in which we describe the mechanisms of AI in detail: Deetz, K., Vogel, H., Knippertz, P., Adler, B., Taylor, J., Coe, H., Bower, K., Haslett, S., Flynn, M., Dorsey, J., Crawford, I., Kottmeier, C., and Vogel, B.: Numerical simulations of aerosol radiative effects and their impact on clouds and atmospheric dynamics over southern West Africa, Atmos. Chem. Phys., 18, 9767–9788, 2018. https://www.atmos-chem-phys.net/18/9767/2018/acp-18-9767-2018.pdf**

(4) P7 L 15-24 : (a) The Aerosol Inflow involves an increase of RH a decrease of temperature but also brings different types of aerosols inland. You discuss the meteorological conditions that have for sure an influence on the ALWC but you never suggest that aerosol components may also have an impact. (b) Section 3.3 (Impact of aerosol modes): First, you should details the different types of aerosols that are predominant during each phase and the mean size distribution associated with each phase. (c) Do

you separate the aerosol modes in term of chemistry within your model? (d) It seems, according to P8 L25, that coarse mode is only made of sea salt particles. (e) How do you take into account dust then? (f) The comparison with chinese field campaigns need to be clarified. Are the different types of aerosols similar in China and Africa? (g) Did Chen et al. (2012) performed their measurements during the monsoon period?

**We have separated this remark in subsections (a)-(g): (a) In a draft version of our manuscript we have had a further subsection that dealt with the effect of the aerosol composition on ALWC during the diurnal evolution. In fact, ACC is dominating in all three phases. In Phase 2 the ALWC contribution from ACC increases because the RH increases. The ALWC contribution from sea salt is generally higher during daytime (although more sea salt is transported inland during night). This is because sea salt also takes up water at RHs that are significantly below 95 % (daytime drying) which is not the case for the submicron particles within ISORROPIA II. Therefore during daytime sea salt has strongest contributions to the total ALWC. Although, the analysis of these aspects are interesting we decided to exclude it from this study for two reasons: - The discussion of the aerosol composition impact on ALWC is strongly dependent on how it is parameterized in the model. E.g. for sea salt we use the parameterization of Lundgren et al. (2013) which can lead to significant different results when using another parameterization. - Furthermore, the analysis of these aspects distract from the actual goal of this study. We wanted to find a relationship between the ALWC and its impact on the radiative transfer in shortwave and longwave. Section 3 is meant as a rather short transfer part that leads the reader to the core topic assessed in Section 4. Additionally it has to be considered that the ISORROPIA is based on the equilibrium solution. This works well in general but can also lead to substantial deviations. Water is a component of this equilibrium and therefore we cannot separately assess the impact of specific aerosol components on the ALWC.**

(b) The manuscript points out that ACC is dominating in all Phases. In Figure E1 we now have added (in addition to the aerosol diameter) the aerosol number concentration

of the different modes (Review-figure-1).

(c) The aerosol treatment in COSMO-ART is described in detail in Vogel et al. (2009). All aerosol species in COSMO-ART (except of pollen and volcanic ash, which are not considered in this study) are allocated to lognormal aerosol modes. But of course the aerosols undergo aerosol chemistry (e.g. deposition of sulfuric acid on soot particles).

(d) No, COSMO-ART considers sea salt, mineral dust and coarse mode anthropogenic particles in the coarse mode. But with respect to ALWC, only sea salt is relevant within the coarse mode. Therefore this study focuses on sea salt in COARSE.

(e) Mineral dust is treated as chemical inert in COSMO-ART. Of course this is a short-coming, because aged mineral dust in the atmosphere can also be subject to ALWC or other chemical reactions. These effects are not considered in COSMO-ART. It has to be considered that in the research period 2-3 July/ 6-7 July mineral dust plays no role in the monsoon layer.

(f) This aspect refers to a remark that came up already in the first review. Therefore I copy my thoughts and the revision at this place: When focusing on the study of Bian et al. (2014), the observations are related to the time period July-August 2009 and focusing on the chinese provinces Shandong, Hebei, Pekin and Tianjin. The climate in this area is in between humid subtropical and humid continental Summers are hot and rainy with temperatures around 24-28 °C in July with the precipitation maximum in summer via influences from the monsoon. A qualitative analysis of Terra Modis satellite images (of course only one overfly per day) revealed that in the 62 d period of July-August 2009 Shandon was fully covered by clouds on 55 d and partly covered by clouds on 7 days. Therefore the weather conditions during the DACCIWA campaign and HaChi campaign are very similar. Both studies focus on the NH summer. Both areas are located in the NH summer monsoon area with high temperatures and are very frequent covered by clouds. The measurement site for the study of Bian et al. (2014) is Wuqing. For this location, Liu et al. (2011) [Figure 3] shows measurements

of temperature and relative humidity for July-August 2009. Temperature variations are between 20 °C and 32 °C. Relative humidity variations are between 40 % (mostly 60%) and 95 %. The latter is similar to what is modeled for southern West Africa (Fig. 3 in our manuscript) and to what was observed in southern West Africa at Save supersite (Kalthoff et al., 2018, Fig. 3).

Wuqing is about 90 km away from the Gulf of Bohai. So also HaChi focuses on the area near the coast. Wuqing is surrounded by large cities (Peking (80 km away, 21.5 million inhabitants, megacity), Langfang (30 km away, 4.4 million inhabitants), Tianjin (40 km away, 15.5 million inhabitants, megacity), Tangshan (100 km away, 7.6 million inhabitants)). Also southern West Africa has several large cities especially near the coast. However, the populations are generally smaller but on the same order of magnitude (Lagos: 13.7 million inhabitants, Abidjan: 5 million inhabitants). Based on MODIS observations, Bian et al. (2014) show that the averaged AOD values are generally above 0.6 in the research area and 0.7 above Wuqing. For the DACCIWA region we found averaged MODIS AOD values of 0.4-0.7, slightly smaller to what was observed in the HaChi region. However, the validity over land is limited because southern West Africa is virtually always covered by clouds, restricting the observations to a few days.

Based on these findings we came to the conclusion that the general meteorological and aerosol conditions are similar for HaChi and DACCIWA and therefore allow a qualitative comparison e.g. of the ALWC values between both sites.

We added the following passage in the conclusions to account for your remark: "HaChi and DACCIWA both focus on the northern hemispheric monsoon season, capture coastal areas that are frequently covered by clouds, have similar temperature and relative humidity conditions (Liu et al., 2011; Kalthoff et al., 2018) as well as similar aerosol loadings (Bian et al. (2014); Deetz et al. (2018a), allowing for a qualitative comparison of modeled ALWC with measurements during HaChi."

(g) The study of Chen et al. (2012) mainly focus on January 2010 so not on the

monsoon period in contrast to Bian et al. (2012) and Liu et al. (2011).

(5) P10 Section 4.2 : In this section, you are using 3 different figures to describe the effect of ALWC on the shortwave, longwave radiations and 2-m temperature. (a) However, I felt like I did not have any explanations on what you observed. As an example, L16-18 'a decrease in SSR can be observed when considering ALWC for ICA and OCA'. Could you explain why you have the same order of magnitude for OCA and ICA (where the RH should be higher)? (b) 'A change in the cloud cover' âGËŸ Š the cloud is disappearing or strengthening? These are examples, but the entire section is written the same way. (c) According to your conclusions it seems that the cloud presence doesn't affect much the effect of ALWC on radiation. Could you provide anywhere in your manuscript the meteorological and aerosol size distribution differences between OCA and ICA?

**We have separated this remark in subsections (a)-(c): (a) We have applied two model realizations, one is the reference run in which the ALWC is considered in the radiative transfer calculations and the other run is the experiment ("No-ALWC") in which the ALWC is neglected in the radiative transfer. As expected, the incomins surface shortwave radiation (SSR) decreases when we consider ALWC in the radiative transfer. The median reduction is -28 W m-2 for the in-cloud area (ICA) and -15 W m-2 outside of clouds. As expected the reduction is higher in clouds because there the RH is higher and therefore the ALWC increases compared to areas outside of clouds. For ICA the reduction is twice as high as for OCA. It has to be considered that the radiative transfer is a two-stream model (just up and down). The intensity of an incoming beam that passes a certain column is reduced in case of ICA by the ALWC in clouds but also by the ALWC below and above the clouds. In case of OCA the light intensity is reduced only by ALWC outside of clouds in the total column. (I) Even in OCA the RH can reach very high values near 100 % and (II) in ICA the clouds mostly will span only a very small fraction of the total vertical column. (III) Most of the path in OCA AND ICA will be cloud free. The aspects (I-III) let deduce that the ALWC surplus from cloudy regions**

can be high but nevertheless the difference in SSR between ICA and OCA will not be extraordinary high.

(b) We revised the corresponding sentence in the manuscript: "The positive values north of 8 åŮe̥ N in Phase 3 are related to a change in cloud cover (more clouds in Reference), which is not a general feature." If you have detect further imprecise statements, please specify.

(c) Figure E1 (see Review-figure-1) now shows the median aerosol number density for the separate modes on 3 July 2016 6 UTC in the lowest 1500m AGL. Furthermore, Review-figure-2 shows boxplots of the wet diameters for areas with a cloud water greater than zero (in clouds, ICA) and for areas with a cloud water equal zero (off clouds, OCA) also on 3 July 2016 6 UTC in the lowest 1500m AGL. For this time the meteorology (differences in temperature and RH) looks as follows: see Table presented in Review-figure-4.

Review-figure-2 is added in the manuscript as Figure E2 and we added the following passage in the end of Section 4.1: "For Reference on 3 July 6 UTC, Figure E2 shows the median wet diameter separated in ICA and OCA for the lowest 1500 m AGL over Ivory Coast, highlighting the effect that submicron particles (Fig. E2a) need a RH near 100 % to significantly grow, whereas sea salt (Fig. E2b) already shows a growth due to ALWC at lower RH values. The median temperature for ICA (OCA) is 20.9°C (21.7°C) and the median RH for ICA (OCA) is 99.9% (93.2%)."

Minor comments : (6) Page 2 L10 : replace natrium by sodium

**We have changed the manuscript accordingly.**

(7) Page 3 L14-16 : I'm not sure I understand this sentence. You claim : " The RH increasingly affects the relationship between the amount of aerosol and the cloud droplet number concentration". I believe that larger RH could involve more or larger cloud droplets. These results are not from 2015: : :

[Figure]

**We removed this passage and the citation because this is less relevant as you have described.**

(8) Page 3 L28 : I believe that there were no GF measured during AMMA.

**We have corrected the citation.**

(9) P5 L4 : Could 5% of the mass concentration of soot particle be defined considerable ?

**We have rephrased this sentence.**

(10) P5 L16-17 : 'Furthermore, : : : process studies'. I do not understand this sentence. The undisturbed moosoon condition favor NLLS presence? Also, NLLS is not defined in the acronym list.

**We removed the "and" in the corresponding sentence. Yes, undisturbed monsoon conditions favor the process studies, because then the conditions are very similar from day to day, making a short simulation period qualitatively respresentative for longer time periods. Undisturbed monsoon conditions also favor NLLS presence because e.g. the passage of an MCS can disturbe the NLLJ and with that the evolution of NLLS. We added NLLS in the acronym list.**

(11) P5 L28 : please remove 'by a a decrease'

**We have changed the manuscript accordingly.**

(12) P9 section 3.4 : Is this AOD within cloud ? Are you talking about interstitial aerosols? Then the clouds are just considered as a vector for RH increase? There are numerous studies that have shown the contribution of the ALWC to the total AOD (Brock et al., 2015 and 2016; Crumeyrolle et al., 2014; Beyersdorf et al., 2016; Orozco et al. 2016; Eck et al. 2014).

**Yes, we consider clouds as areas where the RH maximizes. And here we don't focus on cloud optical thickness but on the radiative effects that come from the ALWC. Yes,**

you can term it interstitial aerosol. It is right that this effect was already analyzed in several former studies. Nevertheless, the topic is still relevant. The weather forecast model COSMO (not the research model COSMO-ART) still does not consider this effect when calculating the radiative transfer. This study is also meant as a motivation for the model developer to consider these effects, especially when they do forecasts in moist tropical regions. We have not stated that our finding about the strong impact of ALWC on AOD is completely new. We just higlighted this finding as a step towards the subsequent analysis of the ALWC - radiation relationship. But we added the following passage in the introduction to consider your remark: "Several studies analyzed the implication of ALWC to AOD (e.g. Brock et al. 2016, Beyersdorf et al. 2016). Brock et al. (2016) combine aircraft observations with a simple model to analyze the sensitivity of the AOD towards meteorological and aerosol properties in the southeastern United States. The results indicate highest (lowest) sensitivities towards RH (dry and wet aerosol refractive index)."

(13) P12 L22 : please remove 'The' : 'on THE one hand'

**"On the one hand ... on the other hand ..." is a fixed term. Please specify if we misunderstood your remark.**

(14) P13 L3 : please replace 'Al affected' - 'Al affects'

**We have changed the manuscript accordingly.**

(15a) Figure 4 : I'm sure this is a typo : ' same ass '

**We have changed the manuscript accordingly.**

(15b) Relative ALWC should be a proxy for the hygroscopicity of aerosols right ? If yes then it needs to be stated somewhere. And you should present mean aerosol size distribution before this figure for the different phases.

**We added the following sentence in the introduction: "The relative ALWC can be seen as a proxy for the hygroscopiciy of an aerosol species." Figure D1 shows the median**

mass concentration of the single aerosol species, Figure E1 (Review-figure-1) presents the median number concentration as well as dry and wet diameters. Furthermore figure E2 (Review-figure-2) is added to separate the analysis of dry and wet diameters to ICA and OCA. We disagree that a further analysis of the aerosol size distribution, now separated in the three AI phases, is appropriate. Adding more and more figures will distract the reader from the main outcomes and also contradicts your remark (1).

(16) Figure 5 : You should add on the different figures 'TOTAL' , 'AIT', 'ACC' and 'COARSE'

**We have adapted the figure accordingly.**

(17) Figure 6 : Could you add the RH on this figure ?

**We have adapted the figure accordingly (see Review-figure-3).**

(18) Figure 9/10/11 : Could you add on the figure ICA and OCA . I'm sure that will also be clearer if there is REF and REF-No_ALWC

**We have adapted the figure accordingly.**
* * *
[Figure]

**Fig. 1.** Boxplots of (a) aerosol number density (cm-3) and (b) dry (red) and wet (blue) aerosol diameters (um) for AIT and ACC and boxplots of (c) aerosol number density (cm-3) and (d) dry and wet diameter.

**(a)**

$AIT_{ICA}$   $AIT_{OCA}$   $ACC_{ICA}$   $ACC_{OCA}$

0.4
0.3
0.2
0.1
0 µm

**(b)**

0.1µm

$COARSE_{1,ICA}$   $COARSE_{1,OCA}$   $COARSE_{2,ICA}$   $COARSE_{2,OCA}$   $COARSE_{3,ICA}$   $COARSE_{3,OCA}$

**Fig. 2.** Boxplots of aerosol wet diameters (um) for (a) AIT and ACC and (b) COARSE, splitted in the three COSMO-ART sea salt modes as median in the lowest 1500 m AGL on 3 July, 6 UTC by separating in ICA/OCA.

**Fig. 3.** Diurnal cycle of the median GF (%) of GF_AIT (red), GF_ACC (green), GF_COARSE (blue) and RH (%) (blue dashed) in the lowest 1500 m AGL over Ivory Coast (7.5° W–3° W, 4–10° N) on 2-3 July.

```
                                          in clouds  |   off clouds
* * *
Temperature °C (median and standard deviation)   20.9  2.0  |   21.7  2.3
RH % (median and standard deviation)            99.99 1.4  |   93.2  4.9
```

**Fig. 4.** Table summarizing the meteorological conditions on 3 July 2016 6 UTC for "in clouds" (ICA) and "off clouds" (OCA) in the lowest 1500m AGL.

---

## Author Comment (AC3) · 4 Aug 2018

Answer to Referee #2b Konrad Deetz 23 July 2018

Dear Referee (Atmospheric Chemistry and Physics),

thank you for your supplementary report from 10 July 2018. We have accounted for the additional comment in the revised manuscript version. Please find our reply (marked with #) in the following.

Sincerely, Konrad Deetz on behalf of all coauthors

Referee comments: (1) It's not clear if the radiative effect of the ALWC is calculated

only from an increase of the aerosol size or if the refractive index is changing with the water uptake ?

**We see your point. We suggest that the radiative effect of the ALWC is a combination of effects from the aerosol size increase and the change in the refractive index. Both effects are considered in COSMO-ART but in the model output we do not disentangle them. Although, this is an interesting aspect, it is beyond the scope of our study, which aims on the general quantification of the impact of ALWC on the radiative transfer. To account for your remark, we added the following paragraph in the conclusion section: "It is expected, that the radiative effect of the ALWC is determined by a combination of the aerosol size increase and the corresponding change of the refractive index. Although it would be interesting to assess the contribution of each process, this is beyond the scope of this work and has to be left for future studies."**

—————————————————

---

## Editor Decision (ED1)

Manuscript: acp-2018-420
Title: Aerosol liquid water content in the moist southern West African monsoon layer and its radiative impact

Dear Dr. Deetz,

I have read through your manuscript and your response to the reviewers. It is clear that you have worked hard to address all of the reviewer concerns. Furthermore, I agree with both reviewers that the manuscript contains novel and useful research and is thus worthy of publication in ACP. There are however, a number of minor concerns remaining that should be addressed before the manuscript is completely ready for publishing. These are listed as below. Please be sure when you upload your responses that you use track changes as this facilitates my evaluation of your response to the authors. Some of the comments below may simply be due to the fact that I couldn't track down your response.

1. Referee #2 raises an important concern regarding the number of figures being utilized in the paper, as well as the related explanations. This is my primary concern regarding the current manuscript. You currently have 26 figures. I understand the authors' response regarding the need for additional figures related to the additional analysis of 6-7 July, but I would encourage you to assess where you can combine figures. For example, you might want to combine those figures that are the same from each case study (panel a and b). This facilitates direct comparison and you can always come back to the second figure later on in the discussion if need be. Furthermore, this then leads to a more cohesive discussion when comparing the output.

2. Referee #1 enquires in point (7) about why OC dominate the aerosol mass composition. You provided an answer in your response but I would include some of this response in the actual text. Other readers are likely to ask the same question.

3. Referee #1 asks about the total column water. Your answer is sufficient and I note that you have a statement to this effect in the manuscript, however, it would be useful to simply include in brackets the actual heights over which you have done this.

4. Both Referee #1 and #2 raised several questions about how various processes or setups are represented in the model. As a modeler myself, I am in the same position as you in that there are extensive documents describing the model physics. However, it is most helpful to readers to include a short description of those settings that are particularly pertinent to the current study. It only needs to be about a paragraph long but it typically makes an enormous difference to the readability of the paper.

5. Referee #2 asks about the RH comparisons in their point 2. As a reader I would have asked the same question. I suggest that you include a statement as to why these radiosondes were used, and why you did not draw on other measurements, similar to your response to this referee. This is important given the central role of RH to this study. Such a description strengthens your study methodology.

6. Referee #2 requested information about the dynamic of the Atlantic inflow. While the mechanisms have been described in your related study (Deetz et al 2018) and you refer to this manuscript appropriately, I think that it would be beneficial to include 2 sentences in the current manuscript summarizing your response to this request.

7. Finally, Referee #2 was seeking for more of a physical explanation and / or interpretation than appears in the manuscript in their point (5). The authors' response is certainly insightful and it would be useful to include more of these details (and any others the authors may think are pertinent) in the manuscript.

I look forward to reading your revised manuscript.

Kind regards,

Sue van den Heever

---

## Author Response (AR2)

**Answer to Co-Editor Susan van den Heever**
**Konrad Deetz**
**28 August 2018**

Dear Prof. van den Heever,

thank you for your report from 22 August 2018. We have accounted for the comments
and suggestions in the revised manuscript version. Please find our replies (marked with #)
to the individual comments in the following.
The previous upload of the revised manuscript (4 August, accounting for the
remarks of the two referees) already contains a marked-up manuscript version (as a part of the
Author's response) highlighting all changes made by us. In this revision iteration the marked-up
manuscript version is part of the "Author's response" (Part 1: a point-by-point response to the
reviews,
Part 2: a list of all relevant changes made in the manuscript, Part 3: a marked-up manuscript
version) as requested by ACP. I'm sorry that it might be hidden to deep in the document.
In the current case, the marked-up manuscript version only refers to the seven points you have
raised.

Sincerely,
Konrad Deetz on behalf of all coauthors

Co-Editor comments:
Dear Dr. Deetz,
I have read through your manuscript and your response to the reviewers. It is clear that you have
worked hard to address all of the reviewer concerns. Furthermore, I agree with both reviewers
that the manuscript contains novel and useful research and is thus worthy of publication in ACP.
There are however, a number of minor concerns remaining that should be addressed before the
manuscript is completely ready for publishing. These are listed as below. Please be sure when
you upload your responses that you use track changes as this facilitates my evaluation of your
response to the authors. Some of the comments below may simply be due to the fact that I
couldn't track down your response.

(1) Referee #2 raises an important concern regarding the number of figures being utilized in the
paper, as well as the related explanations. This is my primary concern regarding the current
manuscript. You currently have 26 figures. I understand the authors' response regarding the
need for additional figures related to the additional analysis of 6-7 July, but I would
encourage you to assess where you can combine figures. For example, you might want to
combine those figures that are the same from each case study (panel a and b). This facilitates
direct comparison and you can always come back to the second figure later on in the
discussion if need be. Furthermore, this then leads to a more cohesive discussion when
comparing the output.

**We agree on that and have changed the manuscript as follows:**
Combining the figures of the 2-3 July/6-7 July studies:
Figure 3  - Figure 19 (Hovmoeller diagram of RH and ALWC)       -> now Figure 3
Figure 9  - Figure 23 (Hovmoeller diagram of SSR)               -> now Figure 9
Figure 10 - Figure 24 (Hovmoeller diagram of SLR)                -> now Figure 10
Figure 11 - Figure 25 (Hovmoeller diagram of 2 m temperature)    -> now Figure 11
Figure 15 - Figure 26 (ALWC radiation relationship)             -> now Figure 14

Combining the two figures dealing with AOD:
Figure 12 (AOD ECDF) - Figure 13 (Hovmoeller diagram of AOD)     -> now Figure 12

This procedure reduced the number of figures from 26 to 20.

(2) Referee #1 enquires in point (7) about why OC dominate the aerosol mass composition. You
provided an answer in your response but I would include some of this response in the actual
text. Other readers are likely to ask the same question.

**We agree on that and have added the following passage in Section 3.3:**
"Also the DACCIWA aircraft observations reveal a general dominance of organic aerosol
(e.g. Haslett et al., 2018). Biomass burning is an important source of organic aerosol
and likely is responsible for the dominance of organic aerosol over Ivory Coast. Based
on the experience we obtained with COSMO-ART during our two month of forecasting the
atmospheric composition (June-July 2016, 28 km grid mesh size), we observed that the biomass
burning plumes over the Gulf of Guinea (coming from central Africa) frequently swash into the
DACCIWA domain. Also for 6-7 July 2016 the organic aerosol is dominating (not shown)."

(3) Referee #1 asks about the total column water. Your answer is sufficient and I note that you
have a statement to this effect in the manuscript, however, it would be useful to simply
include in brackets the actual heights over which you have done this.

**We agree on that and have changed the corresponding sentence in Section 3.4:**
"Figure 7 shows the total water column (full integration of model layer, 30 km)."

(4) Both Referee #1 and #2 raised several questions about how various processes or setups are
represented in the model. As a modeler myself, I am in the same position as you in that there
are extensive documents describing the model physics. However, it is most helpful to readers
to include a short description of those settings that are particularly pertinent to the current
study. It only needs to be about a paragraph long but it typically makes an enormous
difference to the readability of the paper.

**We screened the two referee answers again, collected the corresponding process-related questions**
and explain how we meet the concerns in the manuscript:
(4a) ISORROPIA II does not include fresh soot for calculation. Did the model assume aged soot
is internally mixed with sulfate in the calculation of optical properties and radiative transfer?
**This is already considered in the manuscript (p. 5 ll. 8-11): "Fresh soot is treated separately**
but is also related to an uptake of water, namely via the condensation of sulfuric acid on the particle.

Nevertheless, this contribution is negligibly small, since a soot particle with a mass fraction of
sulfuric acid that exceeds 5 % is shifted from the fresh soot mode to aged (internally mixed) aerosol

treated by ISORROPIA II."
(4b) Where is the fixed SST value from COSMO-ART coming from?
**We added the following passage in Section 4.2: "The fixed SST is coming from the driving model**
ICON. For ICON, the SST fields are derived daily at 0 UTC based on observations. A detailed
description of the  SST handling in COSMO can be found in Schraff and Hess (2012)."
(4c) The authors mentioned before the RH underestimation may suggest the model derived ALWC

value from this case study is the lower bound (Page 6, line 28-29), how it compared with the double counting of aerosol activate in the model, which tends to overestimate the AWLC, and the uncertainty for the corresponding radiation change calculation?

**We added the following passage in Section 3.2: "This is a source of uncertainty for the calculation of ALWC and the radiative transfer and the question can be raised how this potential RH underestimation compares with the double counting of aerosol in the aerosol activation. It has to be considered that the increase in water uptake is most sensitive to RH in the narrow range of RH > 95 % and less sensitive for RH below 95 %. Therefore, potential deviations should not be overrated. The conception in COSMO-ART, not to remove the activated aerosol from the aerosol population, is done by reason. Model tests in the past that remove the aerosol after activation leads to an efficient and unrealistic cleaning of the atmosphere. But conception of not removing the activated aerosol from the aerosol population does not lead to an overestimation of the ALWC. Instead it is the consideration of two different aspects: (a) aerosol that take up water, (b) a cloud droplet or ice crystal that has an aerosol particle inside. The activated aerosol particle is a cloud droplet (or ice crystal) and the radiative interaction is only related to its quality being a cloud droplet (the negligible small aerosol particle and its ALWC is not considered when we talk about the interaction between cloud droplet and radiation). On the other hand we have the aerosol in the aerosol population that can take up water when it is hygroscopic. In this case, there is an interaction between the aerosol particle (combination of aerosol and ALWC) and the radiation. Therefore we expect that we do not per se overestimate the ALWC with our model concept. But of course, we see uncertainties in the corresponding radiative transfer calculations. With our existing model system and the model realizations we have conducted for this study it is not possible to quantify these uncertainties or to set them in relation to the uncertainty that comes from deviations in RH."**

(4d) Do you separate the aerosol modes in term of chemistry within your model?

**All aerosol species in COSMO-ART (except of pollen and volcanic ash, which are not considered in this study) are allocated to lognormal aerosol modes. These modes are the basic classification system of aerosol in COSMO-ART and is therefore also respected in the aerosol chemistry. We consider the deposition of sulfuric acid on soot particles. Fresh soot is in the mode SOOT and is moved to AIT or ACC (depending on the size) when it is aged. The formation of secondary aerosol from the gas phase leads to a surplus of particles in AIT.**

To complete the description of aerosol chemistry in Section 2 we added the following sentence: "Secondary organic aerosol (SOA), formed from the gas phase and being a source for AIT, is treated by the Volatility Basic Set (VBS) approach (Athanasopoulou et al., 2013)."

(4e) It seems, according to P8 L25, that coarse mode is only made of sea salt particles.

**This is already considered in the manuscript (p. 5 ll. 14-15): The coarse mode aerosols of anthropogenic and mineral origin are not related to ALWC in COSMO-ART and therefore also neglected in the following."**

(4f) How do you take into account dust then?

**This is already considered in the manuscript (pp. 4-5 ll. 33-2): "Pure soot as well as the coarse mode of anthropogenic and mineral origin are not considered, since their contribution to ALWC is either not considered in COSMO-ART or the contribution is negligible."**

(5) Referee #2 asks about the RH comparisons in their point 2. As a reader I would have asked the same question. I suggest that you include a statement as to why these radiosondes were used, and why you did not draw on other measurements, similar to your response to this referee. This is important given the central role of RH to this study. Such a description strengthens your study methodology.

**We agree on that and adapted the corresponding passage in Section 3.2:**
"RH is the predominant factor for ALWC. However, the evaluation of modeled RH with

observations is impeded by the lack of horizontal fields of observed RH and sparse radiosounding sites over Ivory Coast. Nevertheless, radiosounding is one of the most accurate measurement techniques for quantifying RH and therefore we used the available radiosounding stations (Lamto and Abidjan; Maranan and Fink, 2016) and suitable dates to evaluate the vertical RH profiles of COSMO-ART.

(6) Referee #2 requested information about the dynamic of the Atlantic inflow. While the mechanisms have been described in your related study (Deetz et al 2018) and you refer to this manuscript appropriately, I think that it would be beneficial to include 2 sentences in the current manuscript summarizing your response to this request.

**We agree on that and added the following passage in Section 3.1:**
"The two counteracting effects "pressure difference" and "turbulence difference" determine the AI front and its propagation. During day the land is subject to stronger heating than the Gulf of Guinea, leading to stronger turbulence over land. The turbulence mixes the horizontal momentum of the monsoon flow vertically, impeding the monsoon flow and establishing a frontal structure near the coast. In the evening, the turbulence over land decreases allowing the pressure difference (land-sea pressure gradient) to overcome the effects from turbulence. The front starts to penetrate inland, transporting the post-frontal air inland. Therefore during night the monsoon flow (directed from ocean to land) overcompensates the land breeze that we would expect in the classical land-sea breeze concept."

(7) Finally, Referee #2 was seeking for more of a physical explanation and / or interpretation than appears in the manuscript in their point (5). The authors' response is certainly insightful and it would be useful to include more of these details (and any others the authors may think are pertinent) in the manuscript.

**We relate this remark to the following aspect of point (5), raised by Referee #2 (since the other two aspects of this point already let to changes in the manuscript):**
"In this section, you are using 3 different figures to describe the effect of ALWC on the shortwave, longwave radiations and 2-m temperature. However, I felt like I did not have any explanations on what you observed. As an example, L16-18 'a decrease in SSR can be observed when considering ALWC for ICA and OCA'. Could you explain why you have the same order of magnitude for OCA and ICA (where the RH should be higher)?"

We agree on that and added the following passage in Section 4.2:
"The question may arise, why the reduction of SSR for ICA with its higher RH is not substantially larger than for OCA. As expected, the SSR decreases when we consider ALWC in the radiative transfer and the reduction is higher in clouds because there the RH is higher and therefore the ALWC increases compared to areas outside of clouds. For ICA the reduction is twice as high as for OCA. It has to be considered that the radiative transfer is a two-stream model (downward and upward). The intensity of an incoming beam that passes a certain column is reduced in case of ICA by the ALWC in clouds but also by the ALWC below and above the clouds. In case of OCA the light intensity is reduced only by ALWC outside of clouds in the total column. It has to be considered that:
(I) Even in OCA the RH can reach very high values near 100 %.
(II) In ICA the clouds mostly will span only a very small fraction of the total vertical column.
(III) Most of the path in OCA and ICA will be cloud free.
The aspects (I-III) lead to the conclusion that the ALWC surplus from cloudy regions can be high but nevertheless the difference in SSR between ICA and OCA will not be extraordinary high."

**List of relevant changes**

1. Number of figures reduced/compressed (from 26 to 20).
2. Transfer more of the knowledge provided in the review replies to the actual manuscript, including the treatment of relevant physical processes in the model, the mechanism of the Atlantic Inflow, the relevance of organic carbon, the verification possibilities of the relative humidity and  the interpretation of the relationship between radiation and aerosol liquid water content.

[revised manuscript text omitted]